# Distributed Influence-Augmented Local Simulators for Parallel MARL in Large Networked Systems

**Miguel Suau**
Delft University of Technology
m.suaudecastro@tudelft.nl

**Jinke He**
Delft University of Technology
j.he-4@tudelft.nl

**Mustafa Mert Çelikok**
Aalto University
mustafa.celikok@aalto.fi

**Matthijs T. J. Spaan**
Delft University of Technology
m.t.j.spaan@tudelft.nl

**Frans A. Oliehoek**
Delft University of Technology
f.a.oliehoek@tudelft.nl

## Abstract

Due to its high sample complexity, simulation is, as of today, critical for the successful application of reinforcement learning. Many real-world problems, however, exhibit overly complex dynamics, making their full-scale simulation computationally slow. In this paper, we show how to factorize large networked systems of many agents into multiple local regions such that we can build separate simulators that run independently and in parallel. To monitor the influence that the different local regions exert on one another, each of these simulators is equipped with a learned model that is periodically trained on real trajectories. Our empirical results reveal that distributing the simulation among different processes not only makes it possible to train large multi-agent systems in just a few hours but also helps mitigate the negative effects of simultaneous learning.[1]

## 1 Introduction

Imagine we have to train a team of agents to control the traffic lights of a very large city, so large that we simply cannot control all traffic lights using a single policy. The first step would be to split the problem into multiple sub-regions. A natural division would be to assign one traffic light to each agent. Then, since the agents act locally, we would limit their observations to contain only local information. This partial observability could affect their optimal policies but would also make each individual decision-making problem more manageable (McCallum, 1995; Dearden and Boutilier, 1997). Moreover, we may also want to reward agents only for what occurs in their local neighborhood such that we reduce the variance of the returns (Spooner et al., 2021) and facilitate credit assignment (Castellini et al., 2020). Finally, we could train all agents together on a big traffic simulator that reproduces the global dynamics. However, if the city is truly large, it could take weeks or even months to optimize their policies. That is assuming training actually converges.

One may argue that, since the agents' observations and rewards are local, we could as well train them on separate simulators that model only the local transition dynamics (i.e. cars moving within each of the sub-regions; van der Pol and Oliehoek 2016). This approach might work if the agents' local transitions are isolated from the rest of the system (Becker et al., 2003), but would probably break when the local regions are coupled. This is because the local simulators would fail to account for the fact that the agents' local regions belong to a larger system and depend on one another. A solution is to model the influence the global system exerts on each local region. Fortunately, this does not necessarily imply modelling the entire system, or else we would just use the global simulator. In

---

[1]Source code is available at `https://github.com/INFLUENCEorg/DIALS`.

36th Conference on Neural Information Processing Systems (NeurIPS 2022).

many scenarios, such as in the traffic problem, even though the local regions may be affected by many external variables (e.g.traffic densities in other parts of the city), they are only directly influenced by a small subset of them (e.g., road segments that connect the intersections with the rest of the city). This subset of variables is known as the influence sources. The theoretical framework of Influence-Based Abstraction (Oliehoek et al., 2021) shows that by monitoring the posterior distribution of the influence sources given the action local state history (ALSH), one can simulate realistic trajectories that match those produced by the global simulator. The resulting simulator, known as the influence-augmented local simulator (IALS), has been proven effective in single agent scenarios when combined with planning (He et al., 2020) and reinforcement learning (RL) algorithms (Suau et al., 2022b).

In this paper, we extend the IBA framework to multi-agent domains. We show how to factorize large networked systems, such as the previous traffic example, into multiple sub-regions so that we can replace the GS by a distributed network of IALSs that can run independently and in parallel. There is one important caveat to this. The IBA framework assumes only a single agent is learning at a time. This assumption is needed to make the influence distributions stationary. This implies that in our case since we want the agents to learn simultaneously, previously computed influence distributions would no longer be valid after the agents update their policies. The naive solution would be to recompute new influence distributions every time any agent updates its policy. However, we argue that this is not only impractical, since recomputing the distributions is not without costs, but also undesirable. The theoretical results in Section 4.1 demonstrate that multiple (similar) joint policies may induce the same influence distributions and that even when they vary a little, they can still elicit the same optimal policies. Further, our insights in Section 4.3 hint that what seems to be a problem at first, may in fact be an advantage since in many situations, maintaining the previous influence distributions implies that the local transitions, although biased, remain stationary.

**Contributions**   The main contributions of this paper are: (1) adapting IBA to multi-agent reinforcement learning (MARL),[2] and demonstrating that simultaneous learning is possible without incurring major computational costs, (2) showing that by distributing the simulation among different processes, we can parallelize training and scale up to systems with many agents, (3) revealing that the non-stationarity issues inherent to MARL are partly mitigated as a result of this training scheme.

## 2   Related Work

A few prior works have investigated the computational benefits of factorizing large systems into independent local regions (Nair et al., 2005; Varakantham et al., 2007; Kumar et al., 2011; Witwicki and Durfee, 2011). Unfortunately, since local regions are often coupled to one another, such factorizations are not always appropriate. Nonetheless, in many cases, the interactions between regions occur through a limited number of variables. Using this property, the theoretical work by Oliehoek et al. (2021) on influence-based abstraction (IBA) describes how to build influence-augmented local simulators (IALS) of local-POMDPs, which model only the variables in the environment that are directly relevant to the agent while monitoring the response of the rest of the system with the influence predictor. The problem is that the exact computation of the conditional influence distribution is intractable, and we can only try to estimate it from data. Congeduti et al. (2021) provide theoretical bounds on the value loss when planning with approximate influence predictors. The work by He et al. (2020) has empirically demonstrated the advantage of this approach to improve the efficiency of online planning in two discrete toy problems. Suau et al. (2022b) scale the method to high-dimensional problems by integrating the IBA framework with single-agent RL showing that the IALS can train policies much faster than the GS. In this paper, we extend the IBA solution to MARL and explain how to build a network of independent IALS such that we can train agents in parallel.

One of the consequences of training agents on independent simulators is that the non-stationarity issues arising from having the agents learn simultaneously are partly mitigated. There is a sizeable body of literature that concentrates on this issue (Hernandez-Leal et al., 2017), we include a review of these works in Appendix B for completeness. However, we note that the main purpose of this paper is to scale MARL up to systems with many agents. Hence, we are not concerned here with comparing our method with those that exclusively target non-stationarity, especially given that, for scalability reasons, these cannot be applied to the high-dimensional problems we consider here.

---

[2]Although the original IBA formulation (Oliehoek et al., 2012) is already framed as multi-agent, it assumes agents learn one at a time while the other agents' policies are fixed.

# 3 Preliminaries

The type of problems we describe in the introduction can be formulated as factored partially observable stochastic games (Hansen et al., 2004), which are defined as follows.

**Definition 1** (fPOSG). A factored partially observable stochastic game (fPOSG) is a tuple $\langle N, S, A, T, \{R_i\}, \Omega, \{O_i\} \rangle$ where $N = \{1, ..., n\}$ is the set of $n$ agents, $S$ is the set of $j$ state variables $S = \{S^1, ..., S^k\}$, such that every state $s^t \in \times_{j=1}^k S^j$ is a $k$-dimensional vector $s^t = \langle s^{1,t}, ..., s^{k,t} \rangle$, $A = \times_{i \in N} A_i$ is the set of joint actions $a^t = \langle a_1^t, ..., a_n^t \rangle$, with $A_i$ being the set of actions for agent $i$, $T$ is the transition function, with $T(s^{t+1}|s^t, a^t)$, $R_i(s^t, a_i^t)$ is the immediate reward for agent $i$, $\Omega = \times_{i \in N} \Omega_i$ is the set of joint observations $o_i^t = \langle o_1^t, ..., o_n^t \rangle$, with $\Omega_i$ being the set of observations for agent $i$, and $O_i$ is the observation function for agent $i$, $O_i(o_i^t|s^t)$.

Solving the fPOSG implies finding the policy $\pi_i$ for each agent $i$ that maximizes the expected return $G^t$; as defined in Sutton and Barto (1998). However, agents receive only partial observations $o_i$ of the true state $s$, which are not necessarily Markovian. Therefore, optimal policies are in general history-dependent in a POSG. Hence, we define the agents' policies $\pi_i(a_i^t|h_i^t)$ as mappings from action-observation histories (AOH), $h_i^t = \langle o_i^1, a_i^1, ..., a_i^{t-1}, o_i^t \rangle$, to probability distributions over actions, such that agent $i$'s optimal policy $\pi_i^*$ is the one that for every AOH $h_i^t$ selects the action with the highest $Q$-value $Q^{\pi_i^*}(h_i^t, a_i^t) = \mathbb{E}[G^t \mid h_i^t, a_i^t, \pi_i^*]$.

Given the structural assumptions we made in the introduction about the agents' only being able to observe and be rewarded for what occurs in their local neighborhood, we can narrow down the problem formulation and work with a specific class of fPOSGs called local-form fPOSGs (Oliehoek et al., 2021), which better encompass the problems we consider here.

**Definition 2** (Local-form fPOSG). A Local-form fPOSG is a fPOSG where $O_i$ and $R_i$ depend only on a subset of $m$ state variables $X_i = \{X_i^1, ..., X_i^m\} \subseteq S$, with $m \leq k$ (number of state variables), and agent $i$'s local states $x_i \in \times_{j=1}^m X_i^j$ being vectors $x_i^t = \langle x_i^{1,t}, ..., x_i^{k,t} \rangle$, such that $O_i(o_i^t|s^t) = \dot{O}_i(o_i^t|x_i^t)$ and $R_i(s^t, a_i^t) = \dot{R}_i(x_i^t, a_i^t)$, where $\dot{O}_i$ and $\dot{R}_i$ are the local observation and reward functions for agent $i$.

## 3.1 Influence-Based Abstraction

We now describe the IBA framework (Oliehoek et al., 2021) which intends to simplify the local-form fPOSG formulation by exploiting its structural properties. The framework assumes that there is a single agent $i$ learning at a time while all other agents' policies $\pi_{-i}$ are fixed. Hence, from the perspective of agent $i$, the problem reduces to a POMDP (Kaelbling et al., 1996) where states are pairs $\langle s^t, h_{-i}^t \rangle$, with $h_{-i}^t$ being the AOHs of all agents but agent $i$ (Nair et al., 2003). Finding a locally optimal solution to the Local-form fPOSG can be approached by 'alternating maximization' or 'coordinate ascent'. That is, sequentially iterating over all agents, possibly multiple times, and solving their respective POMDPs (Nair et al., 2003; Oliehoek and Amato, 2016).

Looking at the definition of Local-form fPOSG, one can argue that, when solving for agent $i$, sampling actions from the policies of all the other agents $\pi_{-i}(a_{-i}^t|h_{-i}^t)$ and simulating the transitions $T(s^{t+1}|s^t, a^t)$ of the full set of state variables is unnecessary, and while doing so is possible in small problems, it might become computationally intractable in large domains with many agents. Instead, we can define a new transition function $\bar{T}_i$ that models only agent $i$'s local state variables $x_i$, $\bar{T}_i(x_i^{t+1}|x_i^t, a_i^t)$. The problem is that $x_i^{t+1}$ may still depend on the other agents' actions $a_{-i}^t$ and the non-local state variables $S \setminus X_i$, which means that $\bar{T}_i$ is not well defined. Fortunately, in many problems, only a fraction of the non-local state variables will *directly influence* agent $i$'s local region.

The diagram on the left of Figure 1 is a Dynamic Bayesian Network (DBN) (Pearl, 1988; Boutilier et al., 1999) describing a particular instance of the transition dynamics for a generic agent $i$ in a local-form fPOSG. Agent $i$'s local region, corresponds to the variables that lie within the red box, $x_i \in X_i = \{X_i^1, X_i^2\}$. The diagram also shows the non-local variables, known as influence sources $u_i \in U_i \subseteq S \setminus X_i$, that influence the local region directly. The three dots on the top indicate that there can be, potentially many, other non-local variables in $S$ affecting the local variables $X_i$. These are denoted by $y_i \in Y_i \subseteq S \setminus X_i \cup U_i$. The diagram also shows that agent $j$ can affect agent $i$'s local region through its actions $a_j \in A_j$. However, both $Y_i$ and $A_j$ can only influence $X_i$ via $U_i$. Hence, given $u_i^t$, $x_i^{t+1}$ is conditionally independent of $y_i^t$ and $a_j^t$, therefore $P(x_i^{t+1}|x_i^t, u_i^t, y_i^t, a_j^t) = P(x_i^{t+1}|x_i^t, u_i^t)$.

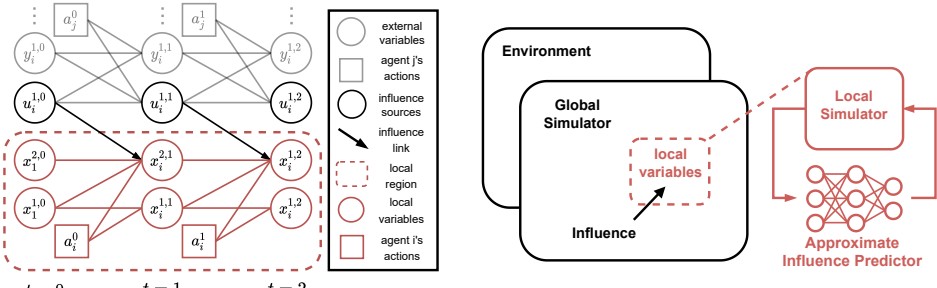

Figure 1: **Left:** A Dynamic Bayesian Network showing agent $i$'s transition dynamics in a local-form fPOSG prototype. **Right:** A conceptual diagram of the IALS.

The above implies that by inferring the value of the influence source $u_i^t$, we can monitor the influence of the other agents and the non-local state variables and thus compute the local state transitions. This can be done by keeping track of the action-local-state history (ALSH) $l_i^t = \langle x_i^1, a_i^1 ..., a_i^{t-1}, x_i^t \rangle$.

**Definition 3** (IALM). An influence-augmented local Model (IALM) for agent $i$ is a tuple $\langle X_i, U_i, A_i, \dot{T}_i, \dot{R}_i, \Omega_i, \dot{O}_i, I_i \rangle$, with local states $x_i \in \times_{j \in |X_i|} X_i^j$, influence sources $u_i \in \times_{j \in |U_i|} U_i^j$, local transition function $\dot{T}_i(x_i^{t+1}|x_i^t, u_i^t, a_i^t)$, local observation function $\dot{O}_i(o_i^{t+1}|x_i^{t+1})$, local reward function $\dot{R}_i(x_i^t, a_i^t)$, and influence distribution $I_i(u_i^t|l_i^t)$.

Using the IALM we can compute agent $i$'s local transitions as

$$P(x_i^{t+1}|l_i^t, a_i^t) = \sum_{u^t} \dot{T}_i(x_i^{t+1}|x_i^t, u_i^t, a_i^t) I_i(u_i^t|l_i^t). \tag{1}$$

Note that, as opposed to the Local-form fPOSG, the transition function $\dot{T}_i$ in the IALM is defined purely in terms of the local state variables and the influence sources. Moreover, since $u_t$ *d-separates* (Bishop, 2006) $x_t$ from $y_t$, we only need to maintain a belief over $u_i^t$, $I_i(u_i^t|l_i^t)$, rather than over the full set of of state variables $s^t$ and other agents' histories $h_{-i}^t$, $P(s^t, h_{-i}^t|l_i^t)$. All in all, this translates into a much more compact, yet *exact* representation of the problem (Oliehoek et al., 2021), which should be computationally much lighter than the original Local-form fPOSG.

### 3.2 Influence-Augmented Local Simulators

Here we briefly describe how the IALM formulation can be used in practice to build IALSs (Suau et al., 2022b), which consist of a *local simulator* and an *approximate influence predictor*.

**Local simulator (LS):** The LS is an abstracted version of the environment that only models a small portion of it. As opposed to a global simulator (GS), which should closely reproduce the dynamics of every state variable, the LS focuses on characterizing the transitions of those variables $X_i$ that agent $i$ directly interacts with, $\dot{T}_i(x_i^{t+1}|x_i^t, u_i^t, a_i^t)$.

**Approximate influence predictor (AIP):** The AIP monitors the interactions between agent $i$'s local region $X_i$, the external variables $Y_i$, and the other agents' actions $A_{-i}$, by estimating $I_i(u_i^t|l_i^t)$. Since, due to combinatorial explosion, computing the exact probability $I_i(u_i^t|l_i^t)$ is generally intractable (Oliehoek et al., 2021), a neural network is used instead to approximate the influence distribution. Thus, we write $\hat{I}_{\theta_i}$ to denote agent $i$'s AIP, where $\theta_i$ are the network parameters. The AIP $\hat{I}_{\theta_i}$ is trained on a dataset $D_i$ of $N$ samples of the form $(l_i^t, u_i^t)$ collected from the GS. Since the role of the AIP is to estimate the conditional probability of the influence sources $u_i^t$ given the past ALSH, we can formulate the task as a classification problem and optimize the network using the expected cross-entropy loss (Bishop, 2006).

## 4 Distributed Influence-Augmented Local Simulators

As mentioned in the previous section, local-form fPOSGs are solved iteratively in the IBA framework. This means that only a single agent can update its policy at a time. Here, we relax this assumption and discuss the advantages and disadvantages of simultaneous learning. Proofs for all the theoretical results in this section can be found in Appendix A.

## 4.1 Enabling Parallelization

The main reason to disallow simultaneous learning is that changes in the other agents' policies can affect agent $i$'s influence distribution $I_i(u_i^t|l_i^t)$, which may become non-stationary. This renders previously computed influences useless because they no longer capture the true response of the global system.

This restriction, however, prevents IBA from unlocking its full potential. The fact that each agent's IALS is independent of the others means that the computations can be distributed among

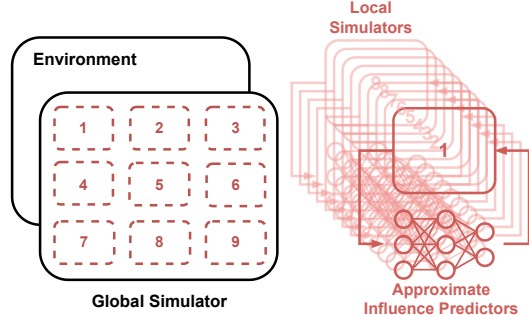

Figure 2: A conceptual diagram of the DIALS

different processes that can run in parallel. Hence, putting aside the non-stationarity issue, and assuming no overhead costs in spawning an increasing number of processes, the total runtime of the method would stay constant if the dimensionality of the global system grew, either because the number of non-local variables or the number of agents increased. This is in contrast to having agents learn simultaneously in the same GS, in which case larger environments imply longer runtimes. Moreover, since each IALS simulates only a portion of the environment the total amount of memory space needed would be split among the different processors. Hence, we could run the simulation on multiple machines with small memory rather than one big machine with very large memory.

In principle, one could prevent the AIPs from becoming stale by simply updating all $\{\hat{I}_{\theta_i}(u_i^t|l_i^t)\}_{i\in N}$ every time any of the other agents changes its policy. However, this creates a difficult moving target problem and makes the whole method very inefficient since, especially in deep RL, policies are updated very frequently. Fortunately, as we argue in the following, in many cases, paying the extra cost of retraining the AIPs is neither necessary nor desirable.

### 4.1.1 Multiple joint policies may induce the same influence distribution

In the following, we show that multiple joint policies may often map onto the same influence distribution $I_i(u_i^t|l_i^t) \in \Psi_i$ for agent $i \in N$.

**Lemma 1.** *Let $\Pi = \times_{i\in N}\Pi_i$ be the product space of joint policies with $\Pi_i$ being the set of policies for agent $i$. Moreover, let $\Psi = \times_{i\in N}\Psi_i$ be the product space of joint influences, with $\Psi_i$ being the set of influence distributions for agent $i$. Every joint policy $\pi \in \Pi$ induces exactly one influence distribution $I_i(u_i^t|l_i^t) \in \Psi_i$ for every agent $i \in N$.*

**Proposition 1.** *The space of joint policies $\Pi = \times_{i\in N}\Pi_i$ is necessarily greater than or equal to the space of joint influences $\Psi = \times_{i\in N}\Psi_i$, $|\Pi| \geq |\Psi|$. Moreover, there exist local-form fPOSGs for which the inequality is strict.*

The advantages of this result were shown empirically by Witwicki and Durfee (2010), who demonstrated that planning times can be reduced by searching the space of joint influences rather than the space of joint policies, which is often much larger. In fact, in the extreme case of local transition independence (Becker et al., 2003),[3] we have that for all joint policies $\pi$ there is a single $\{I_i\}_{i\in N}$

**Corollary 1.** *Let agent $i$'s influence sources $u_i^t$ be independent of the other agents' actions $a_{-i}$. Then, for any joint policy $\pi \in \Pi$, there is a unique influence distribution $I_i^* \in \Psi_i$ for every agent $i \in N$ and $|\Pi| \gg |\Psi| = 1$.*

The result above implies that, in this particular case, we would only need to train the AIPs once at the beginning. Although we do not expect the situation in Corollary 1 to be the norm, we do believe that in many scenarios, such as in the two environments we explore here, we would not need to retrain the AIPs very often because similar joint policies will influence the local regions in very similar, if not in the same, ways. Furthermore, the next result shows that even when this is not the case, an outdated $I_i$ computed from an old joint policy might still produce the same optimal policy for agent $i$.

---

[3]As opposed to IBA, Becker et al. (2003) assume agents are tied by a shared global reward.

#### 4.1.2 Multiple influence distributions may induce the same optimal policy

We use the simulation lemma (Kearns and Singh, 2002) to prove that if two influence distributions are similar enough they will induce the same optimal policy.

**Lemma 2.** *Let $M_i^1$ and $M_i^2$ be two IALMS differing only on their influence distributions $I_i^1(u_i^t|l_i^t)$ and $I_i^2(u_i^t|l_i^t)$. Let $Q_{M_i^1}^{\pi_i}$ and $Q_{M_i^2}^{\pi_i}$ be the value functions induced by $M_i^1$ and $M_i^2$ for the same $\pi_i$. If $I_i^1$ and $I_i^2$ satisfy*

$$\sum_{l_i^t, u_i^t} P(l_i^t|h_i^t) \left| I_i^1(u_i^t|l_i^t) - I_i^2(u_i^t|l_i^t) \right| \leq \xi, \text{ then } \left| Q_{M_i^1}^{\pi_i}(h_i^t, a_i^t) - Q_{M_i^2}^{\pi_i}(h_i^t, a_i^t) \right| \leq \bar{R}\frac{(H-t)(H-t+1)}{2}\xi \tag{2}$$

*for all $\pi_i$, $h_i^t$, and $a_i^t$, where $H$ is the horizon and $\bar{R} = ||R||_\infty$*

Intuitively, Lemma 2 shows that the difference in value between $M_i^1$ and $M_i^2$ is upper-bounded by the maximum difference between $I_i^1$ and $I_i^2$ times a constant. Actually, if the *action-gap* (Farahmand, 2011) (i.e. value difference between the best and the second best action) in one of the IALMs is larger than twice the difference between the $Q_{M_1}^{\pi_i}$ and $Q_{M_2}^{\pi_i}$ the IALMs share the same optimal policy.

**Theorem 1.** *Let $M_i^1$ and $M_i^2$ be two IALMS differing only on their influence distributions $I_i^1(u_i^t|l_i^t)$ and $I_i^2(u_i^t|l_i^t)$. $M_i^1$ and $M_i^2$ induce the same optimal policy $\pi^*$ if, for some $\Delta$,*

$$Q_{M_i^1}^{\pi_i^*}(h_i^t, \bar{a}_i^t) - Q_{M_i^1}^{\pi_i^*}(h_i^t, \hat{a}_i^t) > 2\Delta \quad \forall h_i^t, \hat{a}_i^t \neq \bar{a}_i^t \text{ with } \left| Q_{M_i^1}^{\pi_i}(h_i^t, a_i^t) - Q_{M_i^2}^{\pi_i}(h_i^t, a_i^t) \right| \leq \Delta \quad \forall h_i^t, a_i^t, \pi_i, \tag{3}$$

*where $\bar{a}_i^t = \arg\max_{a_i^t} Q_{M_i^1}^{\pi_i^*}(h_i^t, a_i^t)$*

Combining Lemma 2 and Theorem 1, we see that, because the difference in value between $M_i^1$ an $M_i^2$ depends on $\xi$ (Lemma 2), the closer the distributions $I_i^1$ and $I_i^2$ are, the more likely it is that $M_i^1$ and $M_i^2$ share the same optimal policy. Note that we have no control over the action gap as it is domain-dependent. In some domains, the gap might be large and we can be more relaxed about not retraining the AIPs. In some others, the gap might be small and we may need to retrain the AIPs more frequently.

### 4.2 Algorithm

After the analysis above, we are now ready to present our method, which we call Distributed Influence-Augmented Local Simulators (DIALS). Algorithm 1 describes how we can train multi-agent systems with DIALS. As mentioned earlier, the key advantage of using DIALS is that simulations can be distributed among different processes, and thus training can be fully parallelized. This enables MARL to scale to very large systems with many learning agents. Moreover, following from our theoretical results, AIP training, which can also be done in parallel, is performed only every certain number of timesteps. The hyperparameter $F$ in Algorithm 1 controls the AIPs' training frequency. The effect of $F$ on the learning performance is empirically investigated in Section 5.

---

**Algorithm 1** MARL with DIALS

---

1: Initialize policies $\{\pi_i\}_{i\in N}$ and AIPs $\{\hat{I}_{\theta_i}\}_{i\in N}$
2: **repeat**
3:     Collect datasets $\{D_i\}_{i\in N}$ from GS                                    ▷ See Algorithm 2 in Appendix C
4:     **in parallel, for** $i \in N$ **do**
5:         Train AIP $\hat{I}_{\theta_i}$ on dataset $D_i$                                        ▷ See Section 3.2
6:     **end for**
7:     **in parallel, for** $i \in N$ **do**
8:         **for** $F$ steps **do**                                     ▷ $F$ is the AIPs' training frequency
9:             Simulate trajectories with IALS $\langle \dot{T}_i, \dot{R}_i, \dot{O}_i, \hat{I}_{\theta_i} \rangle$         ▷ See Algorithm 3 in Appendix C
10:             Train policy $\pi_i$                                    ▷ Using any standard RL method
11:         **end for**
12:     **end for**
13: **until** end of training

---

### 4.3 Mitigating the negative effects of simultaneous learning

The results in the previous section showed that the AIPs may not need to be retrained every single time the policies are updated, as the influence distributions may often stay the same or vary only a little. Yet, we now argue that, even when changes in the joint policy do affect the influence distributions $I_i(u_i^t|l_i^t)$ significantly, it may be advantageous not to retrain the AIPs.

First, we have already mentioned that when all agents learn simultaneously in the same simulator the transition dynamics often look non-stationary from the perspective of each individual agent. This may result in sudden performance drops caused by oscillations in the value targets (Claus and Boutilier, 1998). In contrast, when using independent IALS to train our agents, the transition dynamics remain stationary unless we update the AIPs. Hence, by not updating the AIPs too frequently, we get a biased but otherwise more consistent learning signal that the agents can rely on to improve their policies.

Second, we posit that the poor empirical convergence of many off-the-shelf Deep RL methods (Hernandez-Leal et al., 2017; Yu et al., 2021) is also because stochastic gradient descent updates often result in policies that perform worse than the previous ones. Thus, when learning together, agents may try to adapt to other agents' poor performing policies. These policies, however, are likely to be temporary as they are just a result of the inherent stochasticity of the learning process. Similarly, in many environments, agents shall take exploratory actions before they can improve their policies, which may also negatively impact cooperation if they learn simultaneously (Zhang et al., 2009, 2010). In our case, we can again benefit from the fact that the AIPs need to be purposely retrained, and do so only when the policies of the other agents have improved sufficiently.

Even though further theoretical analysis would be needed to be more conclusive about the benefits of using independent simulators, the observations above give reasons to believe that what we initially described as a problem may in fact be an advantage. This view is also supported by our experiments.

## 5 Experiments

The goal of the experiments is to: (1) test whether we can reduce training times by replacing GS with DIALS, (2) investigate how the method scales to large environments with many learning agents, (3) evaluate the convergence benefits of using separate simulators to train agents rather than a single GS, and (4) study the effect of the AIPs' training frequency $F$ on the agents' learning performance.

### 5.1 Experimental setup

Agents are trained independently with PPO (Schulman et al., 2017)[4] on (1) the global simulator (GS), (2) distributed influence-augmented local simulators (DIALS) with AIPs trained periodically on datasets collected from the GS using the most recent joint policy, (3) DIALS with untrained AIPs (untrained-DIALS).

To measure the agent's performance, training is interleaved with periodic evaluations on the GS. The results are averaged over 10 random seeds on all except on the largest scenarios ($10 \times 10$) for which, due to computational limitations we could only run 5 seeds. We report the mean return of all learning agents. We also compare the simulators in terms of total runtime. For DIALS this includes the agents' training time, the AIPs' training time, and the time for data collection.

### 5.2 Environments

**Traffic control** The first domain we consider is a multi-agent variant of the traffic control benchmark proposed by Vinitsky et al. (2018). In this scenario, agents are requested to manage the lights of a big traffic network. Each agent controls a single traffic light and can only observe cars when they are inside the intersection's local neighborhood. Their goal is to maximize the average speed of cars within their respective intersections. To demonstrate the scalability of the method we evaluate DIALS on four different variants of the traffic network with 4, 25, 49, and even 100 intersections (agents). A screenshot of the traffic network with 25 intersections is shown in Appendix F. The GS and LS are built using Flow (MIT License) (Wu et al., 2017) and SUMO (Eclipse Public License Version 2)

---

[4]The vanilla PPO algorithm with decentralized value functions (independent PPO; IPPO) has been shown to perform exceptionally well on several multi-agent environments (de Witt et al., 2020; Yu et al., 2021).

(Lopez et al., 2018). The GS simulates the entire traffic network while each LS models only the local neighborhood of each intersection $i \in N$ (Figure 10 in Appendix F). We use the same LS for every agent-intersection but since, depending on where they are located, they are influenced differently by the rest of the traffic network, we have separate AIPs, $\{I_{\theta_i}\}_{i \in N}$, for each them. These are feedforward neural networks with the same architecture but different weights $\theta_i$, trained periodically with frequency $F$ on datasets $\{D_i\}_{i \in N}$ collected from the GS. The influence sources $u_t^i$ are binary variables indicating whether or not a car will be entering from each of the four incoming lanes.

**Warehouse Commissioning**    The second domain we consider is a warehouse commissioning task (Suau et al., 2022b). A team of robots (blue) needs to fetch the items (yellow) that appear with probability 0.02 on the shelves (dashed black lines) of the warehouse (see Figure 9 in Appendix F). Each robot has been designated a $5 \times 5$ square region and can only collect the items that appear on the shelves at the edges. The regions overlap so that each of the 4 item shelves in a robot's region is shared with one of its 4 neighbors. The robots receive a reward between $[0, 1]$ when collecting an item. The exact value depends on how old the item is compared to the other items in their region. This is to encourage the robots to collect the oldest items first. The robots receive as observations a bitmap encoding their own location and a set of 12 binary variables that indicate whether or not a given item needs to be collected. The robots, however, cannot see the location of the other robots even though all of them are directly or indirectly influencing each other through their actions. We built four variants of the warehouse with 4, 25, 49, and 100 robots (agents). A screenshot of the warehouse with 25 robots is shown in Appendix F. The GS simulates the entire warehouse while the LS models only a $5 \times 5$ square region (Figure 10 in Appendix F). We use the same LS for every robot (agent) but since depending on where they are located they are influenced differently by the rest of the robots, we have separate AIPs, $\{\hat{I}_{\theta_i}\}_{i \in N}$, for each of them. These are GRUs (Cho et al., 2014) with the same architecture but different weights $\theta_i$, which we train periodically with frequency $F$ on datasets $\{D_i\}_{i \in N}$ collected from the GS. Robot $i$'s influence sources $u_i^t$ encode the location of the four neighbor robots. If its AIP $\hat{I}_{\theta_i}$ predicts that any of the neighbor robots is at one of the 12 cells within its region, and there is an active item on that cell, that item is removed and robot $i$ can no longer collect it.

## 5.3    Results

**GS vs. DIALS**    The two plots on the left of Figures 3a and 3b show the average return as a function of the number of timesteps obtained with GS, DIALS, and untrained-DIALS on the 4-agent traffic and warehouse environments. Shaded areas indicate the standard error of the mean. Agents are trained for 4M timesteps on all three simulators. The results reveal that, while agents trained on DIALS seem to converge steadily towards similar high-performing policies, agents trained with the GS often get stuck in local minima, hence the poor mean episodic reward and large standard error obtained with the GS relative to that of the DIALS. In contrast, the low performance of agents trained with the untrained-DIALS indicates that estimating the influences correctly is important for learning good policies. It is worth noting that the gap between the GS and the DIALS is larger in Figure 3b than in Figure 3a. We posit that this is because, in the warehouse domain, agents are more strongly coupled. For comparison, the dashed-black lines in the plots on the left of Figures 3a and 3b show the performance of hand-coded policies. For the traffic domain, we used fixed traffic light controllers that were extensively optimized by Wu et al. (2017). For the warehouse domain, we hand-coded policies that follow the shortest path toward the oldest item in the agent's region. The learning curves for the other scenarios are provided in Appendix D together with further discussion on these results.[6]

**Scalability**    The benefits of parallelization are more apparent when moving to larger environments. The two bar plots in Figures 3a and 3b depict the final average return and the total run time of training 4, 25, 49, and 100 agents on the two tasks for 4M timesteps. The plots show that DIALS scales far better than the GS to larger problem sizes, while also yielding better-performing policies. For example, training **100 agents** on the traffic network takes less than **6 hours** with the DIALS, whereas training them with the GS would take more than **10 days**. This is a **speedup factor of 40**. In fact,

---

[5]A video showing the GS of the traffic network and one of the IALS is provided at `https://youtu.be/DgVE6OIQQz8`.

[6]A video showing how agents trained with DIALS perform on the 100-agent variant of the traffic scenario is provided at `https://youtu.be/G9EthZ-G3vo`.

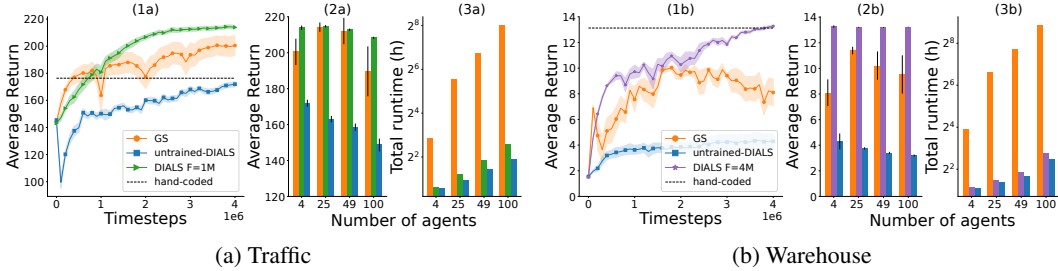

(a) Traffic                                  (b) Warehouse

Figure 3: **(1a) and (1b)** Learning curves with the three simulators on the 4-intersection traffic and 4-robot warehouse environments. **(2a) and (2b):** Final average return of agents trained with the three simulators for 4M timesteps. **(3a) and (3b):** Total runtime of training with the three simulators for 4M timesteps. The $y$-axis is in $\log_2$ scale.

since the maximum execution time allowed by our computer cluster is 1 week, the results reported for the GS in the scenarios of size $10 \times 10$ do not correspond to 4M timesteps but the equivalent of 1 week of training. We would also like to point out that, disregarding the overhead costs associated with multiprocessing, the DIALS runtime should remain constant independently of the problem size. However, to update the AIPs, new samples are collected from the GS, which does increase the runtime. This explains the gap between DIALS and untrained-DIALS. That said, the number of samples needed to update the AIPs (80K for traffic and 10K for warehouse) is significantly lower than the samples needed to train the agents (4M), which is why the runtime difference between GS and DIALS is so large. A table with a breakdown of the runtimes is given in Appendix G.

**AIPs' training frequency** Our first results have already demonstrated that isolating the agents in separate simulators and not updating the AIPs too frequently can be beneficial for convergence. We now further investigate this phenomenon by evaluating the agents' learning performance for different values of the hyperparameter $F$. The two plots on the left of Figures 4a and 4b show the learning curves for agents trained on DIALS where $F$ is set to 100K, 500K, 1M, and 4M timesteps. In the traffic domain, the gap between the green and the purple curve (Figure 4a) suggests that it is important to retrain the AIPs at least every 1M timesteps, such that agents become aware of changes in the other agents' policies. In contrast, in the warehouse domain (Figure 4b), we see that training the AIPs only once at the beginning (DIALS $F = 4$M) seems sufficient. In fact, updating the AIPs too frequently (DIALS $F = 100$K) is detrimental to the agents' performance. This is consistent with our hypothesis in Section 4.3. The plots on the right show the average cross-entropy (CE) loss of the AIPs evaluated on trajectories sampled from the GS. As explained in Section 4 since all agents learn simultaneously, the influence distributions $\{I(u_i^t|l_i^t)\}_{i \in N}$ are non-stationary. For this reason, we see that the CE loss changes as the policies of the other agents are updated. We can also see how the CE loss decreases when the AIPs are retrained, which happens more or less frequently depending on the hyperparameter $F$. Note that the CE not only measures the distance between the two probability distributions but also the absolute entropy. In the warehouse domain, the neighbor robots' locations become more predictable (lower entropy) as their policies improve. This explains why in the first plot from the right the CE loss decreases even though the AIPs are not updated. Also in the same plot, even though by the end of training DIALS $F = 4$M is highly inaccurate, as evidenced by the gap between the purple and the other curves, it is still good enough to train policies that match the performance of those trained with DIALS $F = 500$K and $F = 1$M. This is in line with our results in Section 4. The same plots for the rest of the scenarios are provided in Appendix D.

## 6   Scope and Limitations

DIALS targets networked environments with well-defined local regions where the interactions between different regions occur through a limited number of variables. There is plenty of examples of domains that have this particular structure including traffic, heating and water systems, logistics, telecommunications, etc. Knowledge of the influence sources $U$ and how these affect the local regions is required for building the DIALS. In most cases, however, (as in the two environments we explored here) some domain knowledge suffices to be able to tell what the influence sources are.

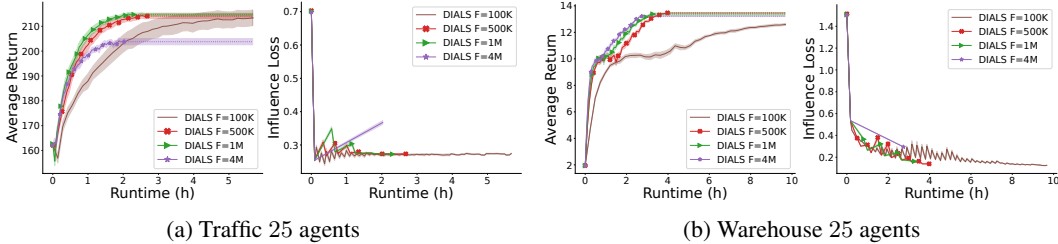

(a) Traffic 25 agents        (b) Warehouse 25 agents

Figure 4: **Left (a) and (b):** Learning curves with DIALS for different values of $F$ on the 25-agent versions of the two environments. **Right (a) and (b):** Influence CE loss as a function of runtime averaged over the 25 AIPs.

Moreover, having or being able to build high-fidelity local simulators of these local regions is also a requirement. Fortunately, there exist plenty of simulators of real systems that can readily be used such as, SUMO (Lopez et al., 2018), Robosuite (Zhu et al., 2020), BRAX (Freeman et al., 2021). There is also a lot of commercial software for building custom-made simulators such as Mujoco (Todorov et al., 2012), or Unity (Juliani et al., 2018). Also, note that most (if not all) of the work that has applied RL to real-world problems relies on simulation to train the policies offline (Bellemare et al., 2020; Degrave et al., 2022). Hence, we believe that DIALS can have a strong impact on many real-world applications.

Finally, although the experiments reveal that DIALS can considerably accelerate training times, it is also memory-demanding. As shown in Appendix H (Table 3), the total memory usage with DIALS increases exponentially with the number of simulators/processes. There is thus a trade-off between fast computation and total memory needed. Note, however, that the memory is split among the different processes. Hence, rather than using a big machine with large memory, DIALS can run on several smaller ones with less memory.

# 7    Conclusion

This paper has offered a practical solution that allows training large networked systems with many agents in just a few hours. We showed how to factorize these systems into multiple sub-regions such that we could build distributed influence-augmented local simulators (DIALS).

The key advantage of DIALS is that simulations can be distributed among different processes, and thus training can be fully parallelized. To account for the interactions between the different sub-regions, the simulators are equipped with approximate influence predictors (AIPs), which are trained periodically on real trajectories sampled from a global simulator (GS). We demonstrated that, although using DIALS agents learn simultaneously, training the AIPs very frequently is neither necessary nor desirable. Our results reveal that DIALS not only enables MARL to scale up but also mitigates the non-stationarity issues of simultaneous learning.

Future work could analyze this phenomenon from a theoretical perspective, study how to adapt DIALS to more strongly coupled domains where frequent training of the AIPs is important, or design a method to directly estimate how the changes in the agents' policies affect the influence distributions. This is so that the AIPs can readily be updated without having to run the GS to generate new samples.

# Acknowledgements

This project received funding from the European Research Council (ERC) under the European Union's Horizon 2020 research and innovation program (grant agreement No. 758824 —INFLUENCE). Mustafa Mert Çelikok is partially funded by KAUTE Foundation - The Finnish Science Foundation for Economics and Technology.

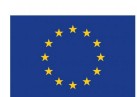 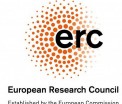

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
