## A  Proofs

**Lemma 1.** *Let $\Pi = \times_{i \in N} \Pi_i$ be the product space of joint policies with $\Pi_i$ being the set of policies for agent $i$. Moreover, let $\Psi = \times_{i \in N} \Psi_i$ be the product space of joint influences, with $\Psi_i$ being the set of influence distributions for agent $i$. Every joint policy $\pi \in \Pi$ induces exactly one influence distribution $I_i(u_i^t | l_i^t) \in \Psi_i$ for every agent $i \in N$.*

*Proof.* We will prove it by contradiction. Let us assume there is a single joint policy $\pi$ that induces two different influence distributions $I_i^1$ and $I_i^2$ on agent $i$. From the definition of influence (Section 4.1; Oliehoek et al. 2021) we have

$$I_i^1(u_i^t | l_i^t) = \sum_{u_i^{t-1}, y_i^{t-1}, a_{-i}^{t-1}} P^1(u_i^t | x_i^{t-1}, u_i^{t-1}, y_i^{t-1}, a^{t-1}) P^1(u_i^{t-1}, y_i^{t-1}, a_{-i}^{t-1} | l_i^t) \tag{4}$$

and

$$I_i^2(u_i^t | l_i^t) = \sum_{u_i^{t-1}, y_i^{t-1}, a_{-i}^{t-1}} P^2(u_i^t | x_i^{t-1}, u_i^{t-1}, y_i^{t-1}, a^{t-1}) P^2(u_i^{t-1}, y_i^{t-1}, a_{-i}^{t-1} | l_i^t). \tag{5}$$

First, we see that, because $\langle x_i^{t-1}, u_i^{t-1}, y_i^{t-1} \rangle$ fully determines the Markov state $s^{t-1}$, the first term in the summation can be computed from the environment's transition function, and thus

$$P^1(u_i^t | x_i^{t-1}, u_i^{t-1}, y_i^{t-1}, a^{t-1}) = P^2(u_i^t | x_i^{t-1}, u_i^{t-1}, y_i^{t-1}, a^{t-1})$$
$$= \sum_{s^t} \mathbb{1}(u^t, s^t) T(s^t | s^{t-1}, a^{t-1}), \tag{6}$$

where $\mathbb{1}(u^t, s^t)$ is an indicator function that determines if the state $s^t$ is feasible in the context of $u^t$.

Further, we know that

$$P^1(u_i^{t-1}, y_i^{t-1}, a_{-i}^{t-1} | l_i^t) = \sum_{h_{-i}^{t-1}} \pi_{-i}(a_{-i}^{t-1} | h_{-i}^{t-1}) P^1(u_i^{t-1} y_i^{t-1}, h_{-i}^{t-1} | l_i^t) \tag{7}$$

$$P^2(u_i^{t-1}, y_i^{t-1}, a_{-i}^{t-1} | l^t) = \sum_{h_{-i}^{t-1}} \pi_{-i}(a_{-i}^{t-1} | h_{-i}^{t-1}) P^2(u_i^{t-1} y_i^{t-1}, h_{-i}^{t-1} | l_i^t) \tag{8}$$

where $P^1(u_i^{t-1} y_i^{t-1}, h_{-i}^{t-1} | l^t)$ and $P^2(u_i^{t-1} y_i^{t-1}, h_{-i}^{t-1} | l^t)$ can be computed recursively as

$$P^1(u_i^{t-1}, y_i^{t-1}, h_{-i}^{t-1} | l_i^t) =$$
$$\sum_{h_{-i}^{t-2}, o_{-i}^{t-1}} O(o_{-i}^{t-1} | x_i^{t-1}, u_i^{t-1}, y_i^{t-1}) \pi_{-i}(a_{-i}^{t-2} | h_{-i}^{t-2}) P^1(u_i^{t-1}, y_i^{t-1}, h_{-i}^{t-2} | l_i^t), \tag{9}$$

and

$$P^2(u_i^{t-1}, y_i^{t-1}, h_{-i}^{t-1} | l_i^t) =$$
$$\sum_{h_{-i}^{t-2}, o_{-i}^{t-1}} O(o_{-i}^{t-1} | x_i^{t-1}, u_i^{t-1}, y_i^{t-1}) \pi_{-i}(a_{-i}^{t-2} | h_{-i}^{t-2}) P^2(u_i^{t-1}, y_i^{t-1}, h_{-i}^{t-2} | l_i^t), \tag{10}$$

with $h_{-i}^{t-1} = \langle h_{-i,t-2}, a_{-i,t-2}, o_{-i}^{t-1} \rangle$. Then, if we further unroll equations (9) and (10) up to timestep 0, we see that all probability distributions in both cases are equivalent and we reach a contradiction. Hence,

$$I_i^1(u_i^t | l_i^t) = I_i^2(u_i^t | l_i^t) \tag{11}$$

$\square$

**Proposition 1.** *The space of joint policies $\Pi = \times_{i \in N} \Pi_i$ is necessarily greater than or equal to the space of joint influences $\Psi = \times_{i \in N} \Psi_i$, $|\Pi| \geq |\Psi|$. Moreover, there exist local-form fPOSGs for which the inequality is strict.*

*Proof.* From Proposition 1 it follows that the space of joint influences $\Psi$ is at most as large as the space of joint policies $\Pi$, $|\Psi| \not> |\Pi|$. Hence, we just need to show that in some cases $\Pi$ is strictly greater than $\Psi$, $|\Pi| > |\Psi|$. $\square$

A clear example is that where each agent's local region $X_i$ is independent of the other agents' policies $\pi_{-i}$ (Becker et al., 2003). That is, the actions of other agents $a_{-i}$ have no effect on agent $i$'s local state transitions. From the definition of IALM (Definition 3) we know that, in our setting, $a_{-i}$ can only affect the local state transitions through $u_i$. Therefore, for the local transitions to be independent the following should hold

$$P(u_i^t | x_i^{t-1}, u_i^{t-1}, y_i^{t-1}, a^{t-1}) = P(u_i^t | x_i^{t-1}, u_i^{t-1}, y_i^{t-1}, a_i^{t-1}) \tag{12}$$

The equation above reflects that only agent $i$ can affect $u_i^t$. Thus, in the event of local transition independence, we have that

$$\forall i \in N : \exists! I_i^*(u_i^t | l_i^t) \in \Psi_i : \forall \pi \in \Pi \left( I_i(u_i^t | l_i^t, \pi) = I_i^*(u_i^t | l_i^t) \right) \tag{13}$$

That is, for any joint policy $\pi \in \Pi$ there is a unique influence distribution $I_i^* \in \Psi_i$ for every agent $i \in N$, and thus, in this particular case, $|\Pi| \gg |\Psi| = 1$.

To prove Lemma 2 we will use the following lemma.

**Lemma 3.** *Let $I_i^1(u_i^t | l_i^t)$ and $I_i^2(u_i^t | l_i^t)$ be two different influence distributions with $M_i^1$ and $M_i^2$ being the IALMs induced by each of them respectively. Moreover, let $P^1(h_i^{t+1} | h_i^t, a_i^t)$ and $P^2(h_i^{t+1} | h_i^t, a_i^t)$ denote the resulting local AOH transitions for $M_i^1$ and $M_i^2$ respectively. The following inequality holds*

$$\sum_{x_i^{t+1}} \left| P^1(h_i^{t+1} | h_i^t, a_i^t) - P^2(h_i^{t+1} | h_i^t, a_i^t) \right| \leq \sum_{l_i^t, u_i^t} P(l_i^t | h_i^t) \left| I^1(u_i^t | l_i^t) - I^2(u_i^t | l_i^t) \right| \quad \forall h_i^t, a_i^t \tag{14}$$

*Proof.*

$$\sum_{h_i^{t+1}} \left| P^1(h_i^{t+1} | h_i^t, a_i^t) - P^2(h_i^{t+1} | h_i^t, a_i^t) \right|$$

$$= \sum_{o_i^{t+1}} \left| \sum_{x_i^{t+1}} O_i(o_i^{t+1} | x_i^{t+1}) \sum_{u_i^t} \dot{T}_i(x_i^{t+1} | x_i^t, u_i^t, a_i^t) \sum_{l_i^t} I^1(u_i^t | l_i^t) P(l_i^t | h_i^t) \right.$$

$$\left. - \sum_{x_i^{t+1}} O_i(o_i^{t+1} | x_i^{t+1}) \sum_{u_i^t} \dot{T}_i(x_i^{t+1} | x_i^t, u_i^t, a_i^t) \sum_{l_i^t} I^2(u_i^t | l_i^t) P(l_i^t | h_i^t) \right|$$

$$= \sum_{o_i^{t+1}} \left| \sum_{x_i^{t+1}} O_i(o_i^{t+1} | x_i^{t+1}) \sum_{u_i^t} \dot{T}_i(x_i^{t+1} | x_i^t, u_i^t, a_i^t) \sum_{l_i^t} P(l_i^t | h_i^t) \left[ I^1(u_i^t | l_i^t) - I^2(u_i^t | l_i^t) \right] \right| \tag{15}$$

$$= \left| \sum_{l_i^t, u_i^t} P(l_i^t | h_i^t) \left[ I^1(u_i^t | l_i^t) - I^2(u_i^t | l_i^t) \right] \right|$$

$$\leq \sum_{l_i^t, u_i^t} P(l_i^t | h_i^t) \left| I^1(u_i^t | l_i^t) - I^2(u_i^t | l_i^t) \right|$$

$\square$

**Lemma 2.** *Let $M_i^1$ and $M_i^2$ be two IALMS differing only on their influence distributions $I_i^1(u_i^t | l_i^t)$ and $I_i^2(u_i^t | l_i^t)$. Let $Q_{M_i^1}^{\pi_i}$ and $Q_{M_i^2}^{\pi_i}$ be the value functions induced by $M_i^1$ and $M_i^2$ for the same $\pi_i$. If $I_i^1$ and $I_i^2$ satisfy*

$$\sum_{l_i^t, u_i^t} P(l_i^t | h_i^t) \left| I_i^1(u_i^t | l_i^t) - I_i^2(u_i^t | l_i^t) \right| \leq \xi, \text{ then } \left| Q_{M_i^1}^{\pi_i}(h_i^t, a_i^t) - Q_{M_i^2}^{\pi_i}(h_i^t, a_i^t) \right| \leq \bar{R} \frac{(H-t)(H-t+1)}{2} \xi$$

$$\tag{2}$$

*for all $\pi_i$, $h_i^t$, and $a_i^t$, where $H$ is the horizon and $\bar{R} = ||R||_\infty$*

*Proof.* This is a special case of the simulation lemma (Kearns and Singh, 2002). We have that the set of local states and actions is the same for both IALMs. Moreover, the reward function is also the same $R^1(x_t, a_t) = R^2(x_t, a_t)$.

$$\left| Q^{\pi_i}_{M^1_i}(h^t_i, a^t_i) - Q^{\pi_i}_{M^2_i}(h^t_i, a^t_i) \right| = \left| \sum_{x^t_i} P(x^t_i|h^t_i)R(x^t_i, a^t_i) \right.$$

$$+ \sum_{h^{t+1}_i, a^{t+1}_i} P^1(h^{t+1}_i|h^t_i, a^t_i)\pi_i(a^{t+1}_i|h^{t+1}_i)Q^{\pi_i}_{M^1_i}(h^{t+1}_i, a^{t+1}_i) - \sum_{x^t_i} P(x^t_i|h^t_i)R(x^t_i, a^t_i) \tag{16}$$

$$\left. - \sum_{h^{t+1}_i, a^{t+1}_i} P^2(h^{t+1}_i|h^t_i, a^t_i)\pi_i(a^{t+1}_i|h^{t+1}_i)Q^{\pi_i}_{M^2_i}(h^{t+1}_i, a^{t+1}_i) \right|,$$

where $P^1(h^{t+1}_i|h^t_i, a^t_i)$ and $P^2(h^{t+1}_i|h^t_i, a^t_i)$ are the AOH transitions induced by $I^1$ and $I^2$ respectively.

$$\left| Q^{\pi_i}_{M^1_i}(h^t_i, a^t_i) - Q^{\pi_i}_{M^2_i}(h^t_i, a^t_i) \right| = \left| \sum_{h^{t+1}_i, a^{t+1}_i} \pi_i(a^{t+1}_i|h^{t+1}_i) \Big[ \right.$$

$$\left. P^1(h^{t+1}_i|h^t_i, a^t_i)Q^{\pi_i}_{M^1_i}(h^{t+1}_i, a^{t+1}_i) - P^2(h^{t+1}_i|h^t_i, a^t_i)Q^{\pi_i}_{M^2_i}(h^{t+1}_i, a^{t+1}_i) \Big] \right|$$

$$= \left| \sum_{h^{t+1}_i, a^{t+1}_i} \pi_i(a^{t+1}_i|h^{t+1}_i) \Big[ \right.$$

$$P^1(h^{t+1}_i|h^t_i, a^t_i)Q^{\pi_i}_{M^1_i}(h^{t+1}_i, a^{t+1}_i) - P^2(h^{t+1}_i|h^t_i, a^t_i)Q^{\pi_i}_{M^1_i}(h^{t+1}_i, a^{t+1}_i)$$

$$\left. + P^2(h^{t+1}_i|h^t_i, a^t_i)Q^{\pi_i}_{M^1_i}(h^{t+1}_i, a^{t+1}_i) - P^2(h^{t+1}_i|h^t_i, a^t_i)Q^{\pi_i}_{M^2_i}(h^{t+1}_i, a^{t+1}_i) \Big] \right|$$

$$\leq \left| \bar{R}(H-t) \sum_{h^{t+1}_i} \left( P^1(h^{t+1}_i|h^t_i, a^t_i) - P^2(h^{t+1}_i|h^t_i, a^t_i) \right) \right.$$

$$\left. + \sum_{h^{t+1}_i, a^{t+1}_i} \pi_i(a^{t+1}_i|h^{t+1}_i)P^2(h^{t+1}_i|h^t_i, a^t_i) \Big[ Q^{\pi_i}_{M^1_i}(h^{t+1}_i, a^{t+1}_i) - Q^{\pi_i}_{M^2_i}(h^{t+1}_i, a^{t+1}_i) \Big] \right| \tag{17}$$

since $Q^{\pi_i}_{M^1_i}(h^{t+1}_i) \leq \bar{R}(H-t)$. Then, from Lemma 3 we know that

$$\sum_{h^{t+1}_i} \left( P^1(h^{t+1}_i|h^t_i, a^t_i) - P^2(h^{t+1}_i|h^t_i, a^t_i) \right) \leq \sum_{l^t_i, u^t_i} P(l^t_i|h^t_i) \left| I^1(u^t_i|l^t_i) - I^2(u^t_i|l^t_i) \right| \leq \xi \quad \forall h^t_i, a^t_i. \tag{18}$$

Hence,

$$\left| Q^{\pi_i}_{M^1_i}(h^t_i, a^t_i) - Q^{\pi_i}_{M^2_i}(h^t_i, a^t_i) \right| \leq \sum_{k=t}^{H} \bar{R}(H-k)\xi = \bar{R}\frac{(H-t)(H-t+1)}{2}\xi. \tag{19}$$

$\square$

**Theorem 1.** *Let $M^1_i$ and $M^2_i$ be two IALMS differing only on their influence distributions $I^1_i(u^t_i|l^t_i)$ and $I^2_i(u^t_i|l^t_i)$. $M^1_i$ and $M^2_i$ induce the same optimal policy $\pi^*$ if, for some $\Delta$,*

$$Q^{\pi^*_i}_{M^1_i}(h^t_i, \bar{a}^t_i) - Q^{\pi^*_i}_{M^1_i}(h^t_i, \hat{a}^t_i) > 2\Delta \quad \forall h^t_i, \hat{a}^t_i \neq \bar{a}^t_i \text{ with } \left| Q^{\pi_i}_{M^1_i}(h^t_i, a^t_i) - Q^{\pi_i}_{M^2_i}(h^t_i, a^t_i) \right| \leq \Delta \quad \forall h^t_i, a^t_i, \pi_i, \tag{3}$$

*where $\bar{a}^t_i = \arg\max_{a^t_i} Q^{\pi^*_i}_{M^1_i}(h^t_i, a^t_i)$*

*Proof.* We will prove it by contradiction. Let us assume there is a policy $\pi^*$ that is optimal for $M^1_i$ but not for $M^2_i$. This implies that, for some $h^t_i$, there is at least one action $\hat{a}^t_i \neq \bar{a}^t_i$ for which

$$Q^{\pi^*}_{M^2_i}(h^t_i, \bar{a}^t_i) < Q^{\pi^*}_{M^2_i}(h^t_i, \hat{a}^t_i) \tag{20}$$

Then, because the maximum gap between $Q_{M_i^1}$ and $Q_{M_i^2}$ is $\Delta$,

$$Q_{M_i^1}^{\pi^*}(h_i^t, \bar{a}_i^t) - \Delta \leq Q_{M_i^2}^{\pi^*}(h_i^t, \bar{a}_i^t) < Q_{M_i^2}^{\pi^*}(h_i^t, \hat{a}_i^t) \leq Q_{M_i^1}^{\pi^*}(h_i^t, \hat{a}_i^t) + \Delta. \tag{21}$$

Therefore, we have

$$Q_{M_i^1}^{\pi^*}(h_i^t, \bar{a}_i^t) - Q_{M_i^1}^{\pi^*}(h_i^t, \hat{a}_i^t) < 2\Delta, \tag{22}$$

which contradicts the statement

$$Q_{M_i^1}^{\pi^*}(h_i^t, \bar{a}_i^t) - Q_{M_i^1}^{\pi^*}(h_i^t, a_i^t) > 2\Delta \quad \forall h_i^t, a_i^t \tag{23}$$

$\square$

# B  Further Related Work

There is a sizeable body of literature that concentrates on the non-stationarity issues arising from having multiple agents learning simultaneously in the same environment (Laurent et al., 2011; Hernandez-Leal et al., 2017). Although oftentimes the problem can be simply ignored with virtually no consequences for the agents' performance (Tan, 1993), in general, disregarding changes in the other agents' policies, and assuming individual Q-values to be stationary, can have a catastrophic effect on convergence (Claus and Boutilier, 1998).

The problem of non-stationarity becomes even more severe in the Dec-POMDP setting (Oliehoek and Amato, 2016) since policy changes may not be immediately evident from each agent's AOH. To compensate for this Raileanu et al. (2018) and Rabinowitz et al. (2018) explicitly train models that predict the other agents' goals and behaviors. In contrast, Foerster et al. (2018a) add an extra term to the learning objective that is meant to predict the other agents' parameter updates. This approach is empirically shown to encourage cooperation in general-sum games. In order to better approximate the value function, several works have studied the use of additional information during training to inform each individual agent of changes in the other agents' policies, leading to the ubiquitous centralized training decentralized execution (CTDE) paradigm. The works by Lowe et al. (2017) and Foerster et al. (2018b) exploit this by training a single centralized critic that takes as input the true state and joint action of all the agents. This critic is then used to update the policies of all agents following the actor-critic policy gradient update (Konda and Tsitsiklis, 1999). Even though the use of additional information to augment the critic may help reduce bias in the value estimates, the idea lacks any theoretical guarantees and has been shown to produce the same policy gradient in expectation as those produced by multiple independent critics (Lyu et al., 2021). Moreover, according to Lyu et al., naively augmenting the critic with all other agents' actions and observations can heavily increase the variance of the policy gradients. Both results, however, assume that the critics have converged to the true on-policy value estimates. The authors do admit that, in practice, critics are often used even when they have not yet converged. In such situations, centralized critics might provide more stable policy updates since they are better equipped to follow the true non-stationary Q values. Following a similar perspective, the concurrent work by Spooner et al. (2021) tries to reduce variance by using a per-agent baseline function that removes from the policy gradient the contributions to the joint value estimates of those agents that are conditionally independent, thus effectively providing the agent with more stable updates. The works by de Witt et al. (2020) and Yu et al. (2021) show that the vanilla PPO algorithm (Schulman et al., 2017) works already quite well on several multi-agent tasks. Yu et al. attribute the positive empirical results to the clipping parameter $\epsilon$, which prevents individual policies from changing drastically, and in turn, reduces the problem of non-stationarity. Li et al. (2021) further analyze this idea and propose a method to estimate the joint policy divergence, which is then used as a constraint in the optimization objective.

# C  Algorithms

The two algorithms below describe how to generate the datasets $\{D_i\}_{i \in N}$ with the GS (Algorithm 2) and how to simulate trajectories with each of the IALS (Algorithm 3).

---

**Algorithm 2** Collect datasets $\{D_i\}_{i \in N}$ with GS

---

**Input:** $T, \{\dot{O}_i\}_{i \in N}, \pi^0 = \{\pi_i^0\}_{i \in N}$ ▷ Global simulator, observation functions, and joint policy
**for** $n \in \langle 0, ..., N/T \rangle$ **do**
    $s^0 \leftarrow$ reset      ▷ Reset initial state
    $\{x_i^0\}_{i \in N} \leftarrow s^0$      ▷ Extract local states from global state
    $\{l_i^0 \leftarrow x_i^0\}_{i \in N}$      ▷ Initialize each agent's ALSH with initial local state
    $\{o_i^0 \sim O_i(\cdot \mid x^0)\}_{i \in N}$      ▷ Sample each agent's observation from $O_i$
    $\{h_i^0 \leftarrow o_i^0\}_{i \in N}$      ▷ Initialize each agent's AOH with initial observation
    **for** $t \in \langle 0, ..., T \rangle$ **do**
        $\{u_i^0\}_{i \in N} \leftarrow s^0$      ▷ Extract each agent's influence sources from global state
        $\{D_i \leftarrow (l_i^t, u_i^t)\}_{i \in N}$      ▷ Append ALSH-influence-source pair to the datasets
        $\{a_i^t \sim \pi(\cdot \mid h_i^t)\}_{i \in N}$      ▷ Sample each agent's action from $\pi_i$
        $s^{t+1} \sim T(\cdot \mid s^t, a^t = \{a_i^t\}_{i \in N})$      ▷ Sample next state from GS
        $\{x_i^{t+1}\}_{i \in N} \leftarrow s^{t+1}$      ▷ Extract local states from global state
        $\{l_i^{t+1} \leftarrow \langle a_i^t, x_i^{t+1} \rangle\}_{i \in N}$      ▷ Append action-local-state pairs to each agent's ALSH
        $\{o_i^{t+1} \sim \dot{O}_i(\cdot \mid x^{t+1})\}_{i \in N}$      ▷ Sample each agent's observation from $\dot{O}_i$
        $\{h_i^{t+1} \leftarrow \langle a_i^t, o_i^{t+1} \rangle\}_{i \in N}$      ▷ Append actions-observation pairs to each agent's AOH
    **end for**
**end for**

---

---

**Algorithm 3** Simulate agent $i$'s trajectory with IALS

---

1: **Input:** $\dot{T}_i, \dot{R}_i, \dot{O}_i, \pi_i, \hat{I}_{\theta_i}$ ▷ local simulator, local reward and observation functions, policy, AIP
2: $x_i^0 \leftarrow$ reset      ▷ Reset initial state
3: $o_i^0 \sim \dot{O}_i(\cdot | x_i^0)$      ▷ Sample observation from $\dot{O}_i$
4: $h_i^0 \leftarrow o_i^0$      ▷ Initialize AOH with initial observation
5: **for** $t \in \langle 0, ..., T \rangle$ **do**
6:     $a_i^t \sim \pi(\cdot \mid h_i^t)$      ▷ Sample action
7:     $\dot{R}_i(x_i^t, a_i^t)$      ▷ Compute reward
8:     $u_i^t \sim \hat{I}_{\theta_i}(\cdot \mid l_i^t)$      ▷ Sample influence sources from AIP
9:     $x_i^{t+1} \sim \dot{T}(\cdot \mid x_i^t, a_i^t, u_i^t)$      ▷ Sample next local state from LS
10:     $l_i^{t+1} \leftarrow \langle a_i^t, x_i^{t+1} \rangle$      ▷ Append action-local-state pair to ALSH
11:     $o_i^{t+1} \sim \dot{O}_i(\cdot \mid x_i^{t+1})$      ▷ Sample observation from $O$
12:     $h_i^{t+1} \leftarrow \langle a_i^t, o_i^{t+1} \rangle$      ▷ Append action-observation pair to AOH
13: **end for**

---

# D   Results

## D.1   DIALS vs GS

The plots in Figures 5 and 6 show the learning curves of agents trained with the GS, DIALS, and untrained-DIALS on the 4 variants of the traffic and warehouse environments (4, 25, 49, and 100 agents). The bar plots show the total runtime of training for 4M timesteps with the three simulators. Shaded areas indicate the standard error of the mean.

The orange curves in Figures 5d and 6d stop at 3.5M and 2M timesteps, respectively. This is because the maximum execution time allowed by our computer cluster is 1 week, and training 100 agents with the GS takes longer. A breakdown of the runtimes for the three simulators is provided in Appendix G. Note that the runtime measurements were made on the only machine in our computer cluster with more than 100 CPUs. This is so that it would fit DIALS when training on the 100-agent variants. However, the experiments that required less than 100 CPUs were ran on different machines with different CPUs.

The bar plots indicate that DIALS is computationally more efficient and scales much better than GS. Note that the $y$ axis is in $\log_2$ scale. Moreover, agents trained with DIALS seem to converge steadily towards similar high-performing policies in both environments, while agents trained with

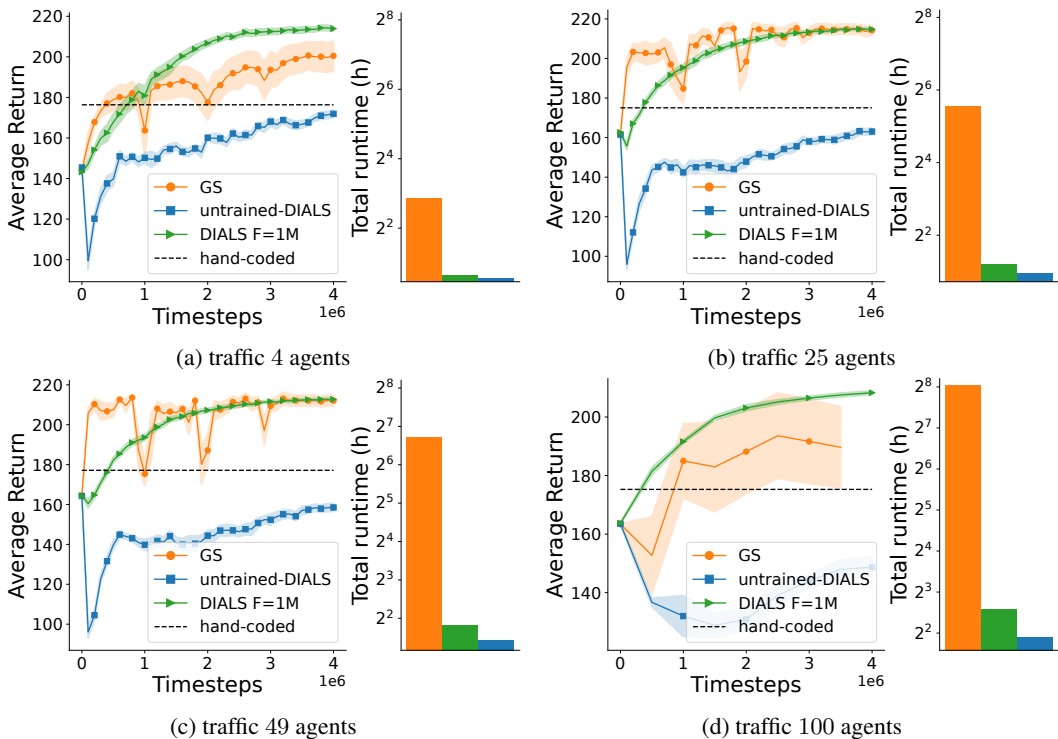

Figure 5: **Left (a), (b), (c), and (d):** Average return as a function of the number of timesteps with GS, DIALS $F = 1$M, and untrained-DIALS on the traffic environment. **Right (a), (b), (c), and (d):** Total runtime of training for 4M timesteps, $y$-axis is in $\log_2$ scale.

the GS suffer frequent performance drops and often get stuck in local minima. This is evidenced by the oscillations in the orange curves, the poor mean episodic reward, and large standard errors compared to the green (traffic) and purple (warehouse) curves. The plots also reveal that estimating the influence distributions correctly is important, as indicated by the large gap between DIALS and untrained-DIALS in both environments.

It is worth noting that the gap between GS and DIALS is larger in the warehouse (Figure 6) than in the traffic environment (Figure 5). We posit that this is because, in the warehouse environment, agents are more strongly coupled. To see this imagine that, by random chance during training, a robot starts favoring items from one shelf over the three others. The robot's neighbors might exploit this and start collecting items from the unattended shelves. However, as soon as this first robot changes its policy and starts collecting items more evenly from all four shelves, the neighbor robots will experience a sudden drop in the value of their policies, which can have catastrophic effects on the learning dynamics. With the DIALS, however, agents are trained on separate simulators and only become aware of changes in the joint policy when the AIPs are retrained. This prevents them from constantly co-adapting to one another. This is in line with our discussion in Section 4.3.

### D.2 AIPs training frequency

The two plots on the left of Figures 7 and 8 show a comparison of the agents' average return as a function of runtime for different values of the AIPs training frequency parameter $F$ (100K, 500K, 1M, and 4M timesteps). For ease of visualization, since DIALS $F = 500$K, $F = 1$M, and $F = 4$M take shorter to finish than DIALS $F = 100$K, the red, green, and purple curves are extended by dotted horizontal lines. Due to computational limitations, we ran these experiments only on the 4, 25, and 49-agent variants of the two environments. We then chose the best-performing values for $F$ ($F = 1$M for traffic and $F = 4$M for warehouse) and used those to run DIALS on the environments with 100 agents.

In the traffic domain, the gap between the green and the purple curve (Figure 7) suggests that it is important to retrain the AIPs at least every 1M timesteps, such that agents become aware of changes

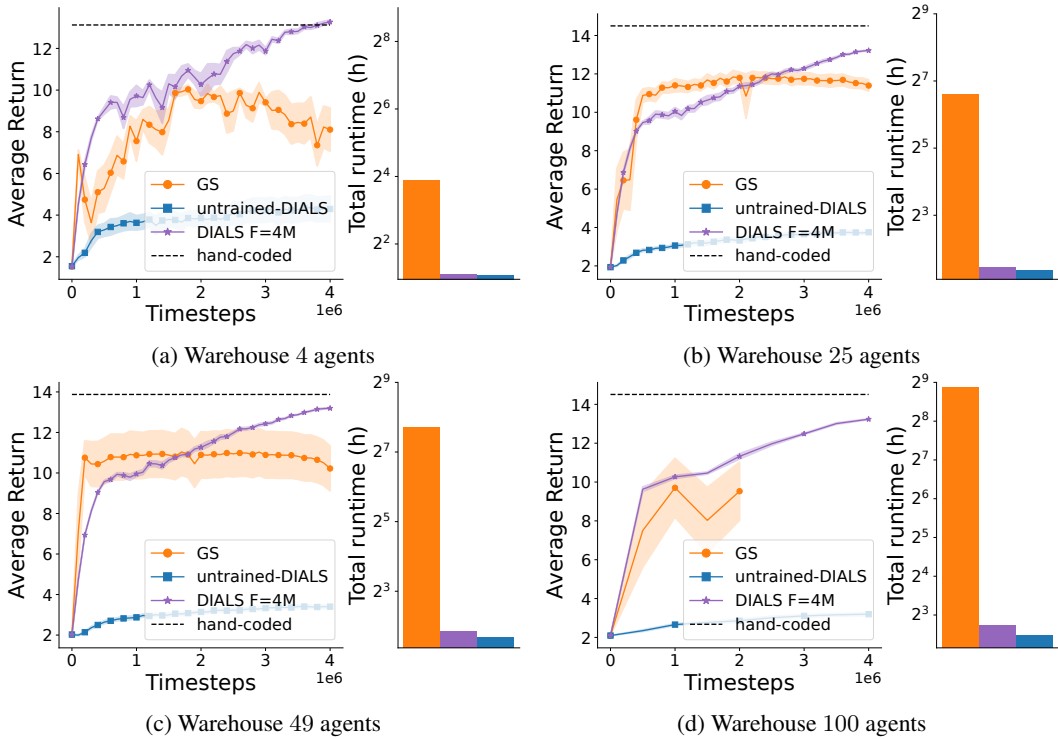

Figure 6: **Left (a), (b), (c), and (d):** Average return as a function of the number of timesteps with GS, DIALS $F = 1$M, and untrained-DIALS on the warehouse environment. **Right (a), (b), (c), and (d):** Total runtime of training for 4M timesteps, $y$-axis is in $\log_2$ scale.

in the other agents' policies. This is consistent on all the three variants (Figures 7a, 7b, and 7c). In contrast, in the warehouse domain (Figure 8), we see that training the AIPs only once at the beginning (DIALS $F = 4$M) is sufficient (Figures 8a, 8b, and 8c). In fact, as indicated by the gap between the brown and the rest of the curves, updating the AIPs too frequently (DIALS $F = 100$K), aside from increasing the runtimes, seems detrimental to the agents' performance. This is consistent with our hypothesis in Section 4.3: "by not updating the AIPs too frequently, we get a biased but otherwise more consistent learning signal that the agents can rely on to improve their policies."

The plots on the right of Figures 7 and 8 show the average cross-entropy (CE) loss of the AIPs evaluated on trajectories sampled from the GS. As explained in Section 4 since all agents learn simultaneously, the influence distributions $\{I(u_i^t|l_i^t)\}_{i \in N}$ are non-stationary. For this reason, we see that the CE loss changes as the policies of the other agents are updated. We can also see how the CE loss decreases when the AIPs are retrained, which happens more or less frequently depending on the hyperparameter $F$. Note that the CE not only measures the distance between the two probability distributions but also the absolute entropy. In the warehouse domain (Figure 8), the neighbor robots' locations become more predictable (lower entropy) as their policies improve. This explains why the CE loss decreases even though the AIPs are not updated. Also note that, in the warehouse environment (Figure 8), even though by the end of training DIALS $F = 4$M is highly inaccurate, as evidenced by the gap between the purple and the other curves, it is still good enough to train policies that match the performance of those trained with DIALS $F = 500$K and $F = 1$M. This is in line with our results in Section 4: "Multiple influence distributions may induce the same optimal policy."

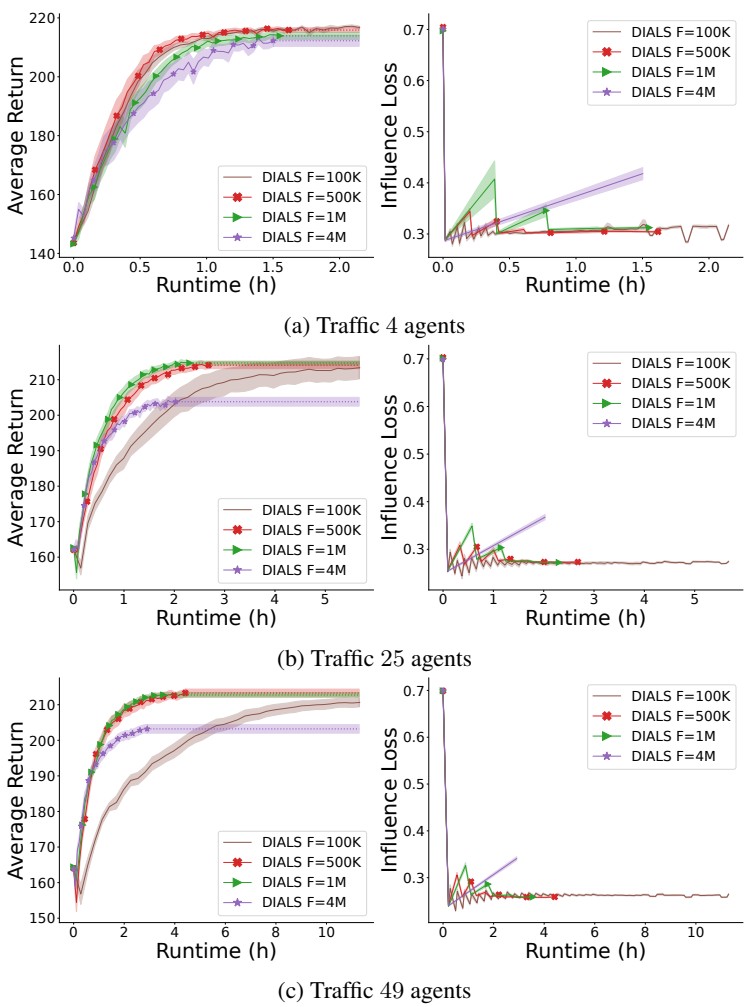

(a) Traffic 4 agents

(b) Traffic 25 agents

(c) Traffic 49 agents

Figure 7: **Left (a), (b), and (c):** Learning curves for different values of $F$ on the 4, 25, and 49 agent versions of the traffic environment. **Right (a), (b), and (c):** CE loss of the AIPs as a function of runtime.

# E Implementation Details

## E.1 Approximate Influence Predictors

Due to the sequential nature of the problem, rather than feeding the full past history every time we make a prediction, we use a recurrent neural network (RNN) (Hochreiter and Schmidhuber, 1997; Cho et al., 2014) and process observations one at a time,

$$P(u_t|l_t) \approx \hat{I}_\theta(u_t|\hat{h}_{t-1}, o_t) = F_{\text{rnn}}(\hat{h}_{t-1}, o_t, u_t), \quad (24)$$

where we use $\hat{h}$ to indicate that the history $h$ is embedded in the RNN's internal memory.

Given that we generally have multiple influence sources $u_t = \langle u_t^1 \dots u_t^M \rangle$, we need to fit $M$ separate models $\hat{I}_{\theta_m}$ to predict each of the $M$ influence sources. In practice, to reduce the computational cost, we can have a single network with a common representation module for all influence sources and output their probability distributions using $M$ separate heads. This representation assumes that the influence sources are independent of one another,

$$I(u_t|l_t) = \prod_{m=0}^{M} P(u_t^m|l_t), \quad (25)$$

which is true for the two domains we study in this paper.

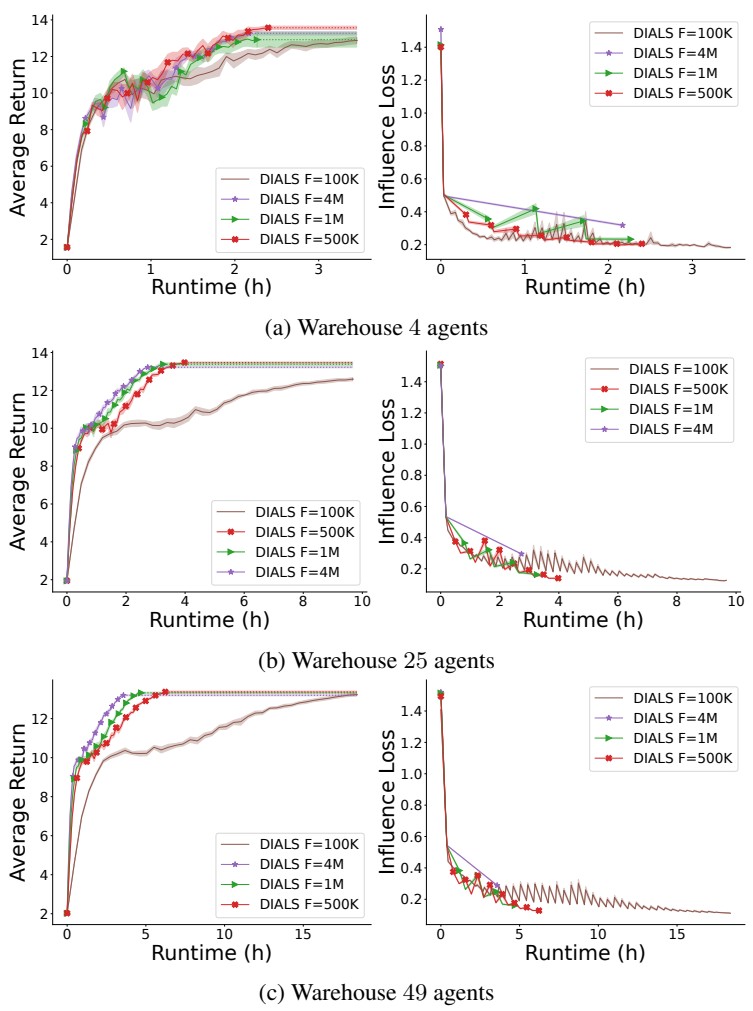

(a) Warehouse 4 agents

(b) Warehouse 25 agents

(c) Warehouse 49 agents

Figure 8: **Left (a), (b), (c), and (d):** Average return as a function of the number of timesteps with GS, DIALS $F = 1M$, and untrained-DIALS on the warehouse environment. **Right (a), (b), (c), and (d):** Total runtime of training for 4M timesteps, $y$-axis is in $\log_2$ scale.

Finally, although according to the POMDP framework we should condition the AIPs on the full AOH, in many domains, one can exploit the structure of the transitions function to find a subset of variables in the AOH that is sufficient to predict the next observation. This subset is known as the d-separating set (Oliehoek et al., 2021), and as shown in Suau et al. (2022a) conditioning the AIPs on this rather than the full AOH can ease the task of approximating the influence distribution.

## E.2 Local regions

When choosing the local regions to build the simulators, the only restriction in terms of size is that these should contain all the necessary information to compute local observations and rewards. In our experiments, we use one simulator per agent since, given that the simulators run in parallel, this is the most computationally efficient way of factorizing the environment. Yet, in certain applications, due to hardware limitations (e.g. not enough CPUs or memory available), it might be necessary to partition the environment into fewer local regions than the number of agents in the environment. Moreover, in some environments (including the two we explore here) better results may be obtained by grouping some of them together in the same simulator. In fact, one could potentially treat the agents in the same group/simulator as a single agent and train a policy to control all of them simultaneously. Note, however, that this is orthogonal to our work as we are mainly concerned with computational speedups.

## F    Simulators

Figure 9 shows two screenshots of the global simulator (GS) for the traffic (left) and warehouse (right) environments with 25 agents each. Figure 10 shows two screenshots of the local simulator (LS) for the traffic (left) and warehouse environments (right). Since all local regions are the same (i.e. $\dot{T}_i$, $\dot{R}_i$, and $\dot{O}_i$ do not change) in the two environments, we use the same LS for all of them. However, because depending on where these are located they are influenced differently by the rest of the system, we train separate AIPs, $\{\hat{I}_{\theta_i}\}_{i \in N}$, for each of them. Note that, we chose the local regions to be the same for simplicity. However, the method can readily be applied to environments with different local transition dynamics $\dot{T}_i$, different local observations $\dot{O}_i$, and/or different local rewards $\dot{R}_i$ for every agent $i \in N$.

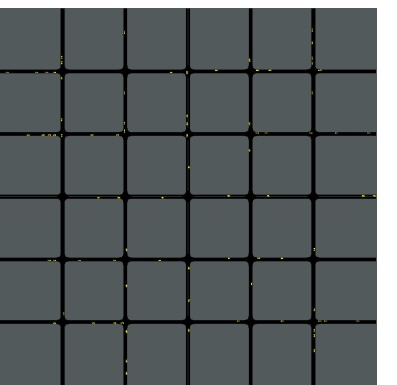 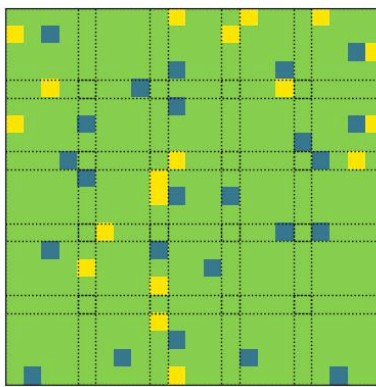

Figure 9: A screenshot of the global simulators for the 25-agent variants of the traffic control (left) and warehouse (right) environments

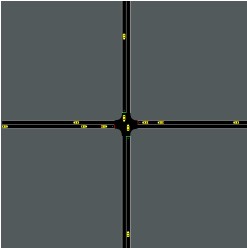 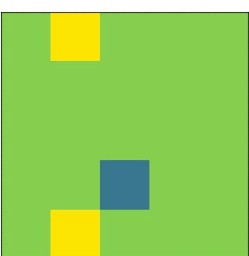

Figure 10: A screenshot of the local simulators for the traffic (left) and warehouse (right) environments. Since all local regions are the same in the two environments, we use the same LS for all of them.

## G    Runtimes

The two tables below show a breakdown of the runtimes for the two environments and the three simulators. These were measured on a machine with 128 CPUs of the type AMD EPYC 7452 32-Core Processor. We used this machine for all our measurements because it is the only one in our computer cluster that can fit DIALS when training on the 100-agent variants of the environments. However, the experiments that required less than 100 CPUs were actually run on different machines.

Table 1: Runtimes for the traffic control environment

| | Agents training (h) | | | | Data collection + influence training (h) | | | | Total (h) | | | |
|---|---|---|---|---|---|---|---|---|---|---|---|---|
| Number of agents | 2 | 25 | 49 | 100 | 2 | 25 | 49 | 100 | 2 | 25 | 49 | 100 |
| GS | 7.24 | 46.96 | 105.41 | 261.06 | - | - | - | - | 7.24 | 46.96 | 105.41 | 261.06 |
| DIALS F=100K | 1.48 | 1.93 | 2.70 | 3.70 | 0.66 | 3.74 | 8.60 | 22.38 | 2.14 | 5.67 | 11.30 | 26.08 |
| DIALS F=500K | 1.48 | 1.93 | 2.70 | 3.70 | 0.13 | 0.75 | 1.72 | 4.48 | 1.61 | 2.68 | 4.42 | 8.18 |
| DIALS F=1M | 1.48 | 1.93 | 2.70 | 3.70 | 0.07 | 0.37 | 0.86 | 2.24 | 1.55 | 2.30 | 3.56 | 5.94 |
| DIALS F=4M | 1.48 | 1.93 | 2.70 | 3.70 | 0.02 | 0.09 | 0.21 | 0.56 | 1.50 | 2.02 | 2.91 | 4.26 |
| untrained-DIALS | 1.48 | 1.93 | 2.70 | 3.70 | - | - | - | - | 1.48 | 1.93 | 2.70 | 3.70 |

Table 2: Runtimes for the warehouse environment

| | Agents training (h) | | | | Data collection + influence training (h) | | | | Total (h) | | | |
|---|---|---|---|---|---|---|---|---|---|---|---|---|
| Number of agents | 2 | 25 | 49 | 100 | 2 | 25 | 49 | 100 | 2 | 25 | 49 | 100 |
| GS | 14.84 | 97.04 | 208.18 | 468.46 | - | - | - | - | 14.84 | 97.04 | 208.18 | 468.46 |
| DIALS F=100K | 2.13 | 2.56 | 3.19 | 5.55 | 1.32 | 7.11 | 15.19 | 45.45 | 4.45 | 9.67 | 18.38 | 51.00 |
| DIALS F=500K | 2.13 | 2.56 | 3.19 | 5.55 | 0.26 | 1.42 | 3.04 | 9.09 | 2.39 | 3.98 | 6.23 | 14.64 |
| DIALS F=1M | 2.13 | 2.56 | 3.19 | 5.55 | 0.13 | 0.71 | 1.52 | 4.54 | 2.26 | 3.27 | 4.71 | 10.09 |
| DIALS F=4M | 2.13 | 2.56 | 3.19 | 5.55 | 0.03 | 0.18 | 0.38 | 1.13 | 2.16 | 2.74 | 3.57 | 6.68 |
| untrained-DIALS | 2.13 | 2.56 | 3.19 | 5.55 | - | - | - | - | 2.13 | 2.56 | 3.19 | 5.55 |

## H  Memory Usage

The table below shows the peak memory usage of the GS and the DIALS. For the latter we provide the memory usage per process and in total. The memory needed for the GS seems to grow logarithmically with the number of agents, whereas for DIALS the memory usage per process stays relatively constant. However, the total amount of memory needed to run DIALS (aggregate of all processes) increases linearly with the number of agents and is considerably larger than that of the GS.

Table 3: Peak Memory Usage in Megabytes (MB)

| Environment | | Traffic | | | | Warehouse | | | |
|---|---|---|---|---|---|---|---|---|---|
| Number of agents | | 4 | 25 | 49 | 100 | 4 | 25 | 49 | 100 |
| GS | | 375.3 | 392.7 | 412.5 | 457.4 | 339.3 | 391.8 | 469.6 | 607.4 |
| DIALS | Per process | 219.5 | 221.0 | 225.8 | 228.7 | 195.6 | 201.9 | 203.7 | 207.5 |
| | Total | 878.0 | 5525.0 | 11064.2 | 22870.0 | 782.4 | 5047.5 | 9981.3 | 20750.0 |

## I  Hyperparameters

The hyperparameters used for the AIPs are reported in Table 4. Since feeding past local states did not seem to improve the performance of the AIPs in the traffic environment we modeled them with FNNs. In contrast, adding the past ALSHs does decrease the CE loss in the warehouse environment, and thus we used GRUs (Cho et al., 2014) instead. The size of the networks was chosen as a compromise between low CE loss and computational efficiency. On the one hand, we need accurate AIPs to properly capture the influence distributions. On the other, we also want them to be small enough such that we can make fast predictions. The hyperparameter named seq. length determines the number of timesteps the GRU is backpropagated. This was chosen to be equal to the horizon such that episodes did not have to be truncated. The rest of the hyperparameters in Table 4, which refer to the training setup for the AIPs, were manually tuned.

Table 4: Hyperparameters for approximate influence predictors (AIPs).

|  | Architecture | Num. layers | Num. neurons | Seq. length | Learning rate | Dataset size | Batch size | Num. epochs |
|---|---|---|---|---|---|---|---|---|
| Traffic | FNN | 2 | 128 and 128 | - | 1e−4 | 1e4 | 128 | 100 |
| Warehouse | GRU | 2 | 64 and 64 | 100 | 1e−4 | 1e4 | 32 | 300 |

The hyperparmeters used for the policy networks are given in Table 5. We chose again GRUs for the warehouse environment and FNNs for the traffic domain, since feeding the previous AOHs did not seem to improve the agents' performance in the latter. The network size and the sequence length parameter for the GRUs were manually tuned on the smallest scenarios with 4 agents.

Table 5: Hyperparmeters for policy networks.

|  | Architecture | Num. layers | Num. neurons | Seq. length |
|---|---|---|---|---|
| Traffic | FNN | 2 | 256 and 128 | - |
| Warehouse | GRU | 2 | 256 and 128 | 8 |

As for the hyperparameters specific to PPO (Table 6), we used the same values reported by (Schulman et al., 2017), and only tuned the parameter $T$, which depends on the rewards and the episode length. $T$ determines for how many timesteps the value function is rollout before computing the value estimates.

Table 6: PPO hyperparameters.

| | |
|---|---|
| Rollout steps $T$ | 16 traffic and 8 warehouse |
| Learning rate | 2.5e-4 |
| Discount $\gamma$ | 0.99 |
| GAE $\lambda$ | 0.95 |
| Memory size | 128 |
| Batch size | 32 |
| Num. epoch | 3 |
| Entropy $\beta$ | 1.0e-2 |
| Clip $\epsilon$ | 0.1 |
| Value coeff. $c_1$ | 1 |