# OpenReview forum: "Distributed Influence-Augmented Local Simulators for Parallel MARL in Large Networked Systems"
_NeurIPS.cc/2022/Conference — NeurIPS 2022 Accept_

### Official Review · Reviewer_iVuf · 2022-07-08

**Rating:** 7
**Confidence:** 3
**Soundness:** 3 good
**Presentation:** 3 good
**Contribution:** 3 good

**Summary:**

This paper extends the influence-based abstraction (IBA) framework to the multi-agent reinforcement learning case (MARL). In the proposed approach, the global system simulator is decomposed into a distributed network of local simulators, each including a set of "influence" variables that model how the local simulators interact with each other. The idea is to independently train agents for each local simulator, in parallel, and periodically update approximate influence predictors (AIP) to account for changes in the other agents' simultaneously trained policies. The paper offers a bit of theoretical analysis justifying the choice of periodically training AIPs. The proposed method (DIALS) is evaluated in two multi-agent environments, showing that DIALS results in not only faster training time compared to training on the global simulator, but also better performance. The authors further numerically study the effect of the frequency of training AIPs, showing that re-training them occasionally can improve performance, but doing it too often can also be detrimental.

**Questions:**

**Main questions:**
- My main question is about the constraints on the system simulator. Am I correct in my understanding that the ability to simulate local regions of the environment, one assigned to every agent, is a constraint placed on the system to be solved? (including freely resetting the local states). Can you discuss this during rebuttal, either explaining why my understanding is wrong, or why this is not too problematic in practice.
- In the paragraph after Theorem 1 (line 236) it is said that one can chose $\Delta$ according to Lemma 2 so guarantee that the conditions of Theorem 1 are met. This is clearly the case for the second condition relating the Q-values obtained with M1 and M2. However, to me this is not obviously true for the first condition of the Theorem, which requires the action-gap in M1 to be larger than $2\Delta$. Am I missing something?
- I would have liked to see more discussion about why training the global simulator (GS) leads to worse performance than using DIALS, since the training data resulting from GS is more accurate. Is this a model capacity issue? Could it be related to the choice of IPPO being better suited to learn with local simulators? Do you observe this when the base MARL method is different?

**Some minor comments/questions:**
  - The paragraph in line 257 discusses the "poor empirical convergence of many Deep MARL methods", but offers very little support for this claim in terms of references or particular experimental results this is referring to. Strong claims like this should be better motivated and discussed, in my opinion.
  - The discussion about RNNs in the AIP paragraph that starts in line 157 seems somewhat out of place. At this point in the paper the discussion has been mostly theoretical, and it seems that all you require here is some generic estimator for the influence distribution. The use of RNN is just one specific way to do this, no? This might be better suited to be described as mostly a technical choice for the experiments.
  - I didn't see a definition for $F$ in Definition 1 (line 88).
  - The claim "In this paper we extend the IBA framework to multi-agent domains" seems overly strong. I took a look at the original IBA paper, and the framework is already tailored to multi-agent problems, so maybe you are referring to the reinforcement learning case? I think that some qualification is missing here.

**Limitations:**

Not much discussion about limitations, including some I outlined above.

**Strengths And Weaknesses:**

**Strengths:**

- *Originality:* I'm not too familiar with the literature of this area, but the proposed extension of IBA to MARL appears to be novel, judging from recent citations to the original IBA paper. As far as I understand, the paper offers a few novel technical contributions on top of the existing IBA theoretical framework, namely:
  - Use parallel training of local simulators, each with their own single-agent policy.
  - Include periodically learned AIPs to account for non-stationarity due to other agents, with the retraining frequency as a hyperparameter controlling addressing non-stationarity vs stability.
  - Proving Theorem 1, which justifies the above choice by essentially implying that one does not need exact influences to obtain optimal policies.
- *Significance:* This is clearly a relevant contribution to the MARL literature, which is an active area of research. The experimental results show that this technique can be used to accelerate and improve the learning of multi-agent policies, and the underlying ideas are simple enough that I could see them being adopted by the community.
- *Clarity:* The paper is reasonably well-written and easy to follow.
- *Quality:* The main ideas are sensible and follow naturally from the IBA framework. I haven't checked the derivations, but the resulting claims are not surprising and appear correct.

**Weaknesses:**
- As far as I can tell, applying DIALS requires one to be able to easily decompose the environment simulator into local pieces that can be executed in parallel. This seems like not only a theoretical limitation, but also a strong practical one, as it means that the environment's simulator must be implemented in a very particular way. I might be missing something, but it doesn't seem like DIALS is a method that can be applied out-of-the-box to any relevant multi-agent environment; it puts strong constraints on how the system is simulated. For example, I'm assuming the environments they used in their experiments were implemented by the authors; at least the wording in line 291 seems to imply this is the case for the traffic control experiments, but nothing is said for the warehouse environment. More generally, I didn't see a lot of discussion of this point in the paper, which I would argue is an important omission.
- The experimental evaluation is somewhat limited, with only two domains and one baseline being considered. Some of the results are not necessarily intuitive and not a lot of discussion is offered (see my last main question below).

---

> ### Author Response · Authors · 2022-08-02
> **Response**
>
> We thank the reviewer for their valuable and constructive feedback, which has helped us improve the paper. We have fixed the typos and incorporated your suggestions in the new revision. Please find the answers to your concerns and specific questions below.
>
> __**Constraints and requirements on the system simulator - "DIALS cannot be applied out-of-the-box to any relevant multi-agent environment":**__
>
> The paper does not claim that DIALS can be applied to any multi-agent environment, only to networked environments with well-defined local regions where the interactions between different regions occur through a limited number of variables. There is plenty of examples of domains that have this particular structure including, traffic networks, heating and water systems, logistics, telecommunication networks, etc. The problem is that there are very few benchmarks in the MARL literature exploring these domains. Investigating how to apply similar ideas to more coupled problems where the local regions are less evident is an interesting direction for future work.
>
> We have added a new section in the appendix (Appendix C)  where we discuss the scope and limitations of the approach.
>
> __**"The environments were implemented by the authors":**__
>
> The traffic control environment was introduced by Vinitsky et al. (2018). The warehouse environment was introduced by Suau et al., 2022. We only varied the size of the environments i.e number of intersections, and robots. These two papers are referenced in Section 5.2.
>
> **Limited experimental evaluation: only two environments and one baseline:**
>
> We respectfully disagree. We have extensively evaluated our method in two environments and four different variants. Unfortunately, no other MARL benchmarks address the type of problem we consider here. That is, simulation of large-scale network systems.
>
> We are comparing DIALS against the global simulator and untrainded-DIALS. The first is meant to evaluate computational speedups,  and the second is used to demonstrate that the regions are coupled and hence that accurate influence predictors are needed. Note that since what we propose is to decompose a problem and build separate simulators, there are simply no other baselines we can compare against.
>
> **Theorem 1:**
>
> The reviewer's understanding is correct. The ‘action gap’ (first condition) is independent of the difference in value between M1 and M2. When we say that $\Delta$ can be chosen according to Lemma 2, we are just trying to tie up the two results (Lemma 2 and Theorem 1). We want to point out that the difference in value is determined by the distance between the influence distributions (Lemma 2). Hence, the closer the two influence distributions are the more likely it is that the two IALMS share the same optimal policy (Theorem 1).
>
> We understand that the wording was confusing and have updated this paragraph in the revision. Moreover, if the paper is accepted we will use part of the extra page to extend the discussion of this result.
>
> **Discussion of the results: why training on the global simulator (GS) leads to worse performance than using DIALS:**
>
> We provide extensive discussion on why we think this is the case in Section 4.3. The reviewer is right that the GS is more accurate and thus in principle, it should yield better results. However, we attribute the performance gap between the GS and DIALS to the non-stationarity issues that appear when agents learn simultaneously.
>
> There is no reason why IPPO would be better suited for local simulators. Agents receive the same observations independently of the simulator being used.  IPPO has been shown to perform very well for MARL (de Witt et al., 2020; Yu et al., 2021). Other independent single-agent learning algorithms will also suffer from the non-stationarity issues inherent to simultaneous learning.
>
> Note that our main contribution is to show that we can speed up the MARL process by decomposing the system in many local regions and distributing the simulation among different processes. The better convergence properties of DIALS compared to GS was just a fortunate consequence of this decomposition. That’s why we don’t compare with other MARL methods that explicitly target non-stationarity, which in any case, due to scalability reasons, could not be applied to the high-dimensional problems we consider here.

---

> > ### Comment · Reviewer_iVuf · 2022-08-09
> > **Thanks for including Appendix C**
> >
> > I appreciate the discussion in Appendix C regarding the scope and limitations of the simulator, I think it adds more clarity to the paper. I will increase my score to 7, mainly because the authors' response regarding the limited baselines in the experiment is convincing to me, but also because I think the limitations placed on the simulator are not as big of a negative point as my initial review made it seems to be.
> >
> > That being said, I still think these limitations should be emphasized in the *main* body of the paper (rather than the appendix), and more details should be added about how the authors were able to do this for their experiments (this could go in the appendix). Their rebuttal makes it seem as if they just took these environments, changed some parameters and ran it, but if that's the case, then how were you "able to build high-fidelity local simulators of these local regions" (to quote Appendix C)? Did the original implementations of these environments already provide these local simulators? I'm assuming that most out-of-the-box simulators would just let you run a global simulation, so you would have to modify the environment's implementation to run networked local simulators in parallel, but I might be wrong here.

---

### Official Review · Reviewer_EtVj · 2022-07-10

**Rating:** 8
**Confidence:** 4
**Soundness:** 4 excellent
**Presentation:** 4 excellent
**Contribution:** 4 excellent

**Summary:**

The authors extend the influence augmented local simulator paradigm to the multi-agent case and show that doing this leads to improvement in both run-time and training outcomes.

**Questions:**

### Questions:

- Is it necessary to group every agent into its own simulator? Could the result be better if instead you grouped them into sets of 2x2? I’d love to see this experiment (though, note, it will not affect my review score I just think it’ll make for a better paper!)
- I’m curious about the non-improvement using recurrent neural networks; is it possible that the RNNs are simply better able to “solve” their local environment and this causes failure in the global simulator due to overfitting to the local simulator? Is it necessary for the agents to not do too well in the local environment?
- Is the time-index available to the influence estimators?
- How many rollouts are in the training batch for PPO? This seems to be missing from the hyperparameter section.

**Limitations:**

It is not fully clear from the authors how much work went into tuning the simulation scheme and consequently it is not fully clear how robust the scheme is. This would be useful information to include in the appendix.

**Strengths And Weaknesses:**

Strengths:
- Improvements in runtime
- Improvements in agent return
Weaknesses:
- Theorem bounds are very loose and consequently don't actually provide much intuition

### Writing:

- The caption in Figure 3 is quite hard to follow. It’s not clear what “first (a) (b)” is nor what the “third a and b” would be
- In the definition of the fPOSG, it seems F is not defined?
- Why does Figure 3 use the 4M agent when it has significantly worse return than some of the more frequent updating agents i.e. the 1M or 500K ones
- It’s probably too late to change this but note that DIAL is already the name of a method in the MARL literature ([https://proceedings.neurips.cc/paper/2016/file/c7635bfd99248a2cdef8249ef7bfbef4-Paper.pdf](https://proceedings.neurips.cc/paper/2016/file/c7635bfd99248a2cdef8249ef7bfbef4-Paper.pdf)). Up to you what you do with this information but I figured it was good for you to be aware of this.

### Correctness

- Lines 116-118 contain a claim about being able to solve local form fPOSG by sequential iteration over agents and solving their POMDP. If this is true, it probably requires a reference or a pointer to the appendix; it is not obviously true from simply reading it.

### Experiments:

- Readers of this paper likely do not have intuition about the meaning of the reward in the traffic light and warehouse case. If given sufficient time, it might be worth including some metrics like speed or objects collected. Similarly, it might be useful to include a hand-tuned baseline (for example, in the traffic light case a fixed set of timing signals) to give a sense of whether the RL policy is learning anything good or is still underperforming simple baselines. Including these experiments or results will not improve my score but I do think it’ll make for a better paper!

---

> ### Author Response · Authors · 2022-08-02
> **Response**
>
> We thank the reviewer for their valuable and constructive feedback, which has helped us improve the paper. We have fixed the mistakes and typos and have updated the paper according to your suggestions. Please find the answers to your specific questions below.
>
> **Theorem bounds and intuition:**
>
> The intuition we wanted to bring forward is that the closer the influence distributions are, the more likely they are to induce the same optimal policies. Our goal is not to provide tight bounds but to support this claim. We have clarified this in the paper and,  if the paper is accepted, will use part of the extra page to extend the discussion of this result.
>
> Theorem 1 shows that two IALMs that differ only on their influence distributions share the same optimal policy if the ‘action gap’ is larger than twice their difference in value. Moreover, we know that the difference in value depends on the distance between the two influence distributions (Lemma 2). This implies that during training as long as the influence distributions do not change significantly, the influence predictors do not need to be retrained, which in practice means that we can further reduce the AIPs’ training frequency.
>
> **Reward meaning and hand-coded baselines:**
>
> The reward in the traffic light domain is the average speed of cars at every timestep divided by the maximum allowed speed. Hence the reported average return is a proxy for the average speed. In the warehouse domain, the return is the number of items collected weighted by how old each item was compared to other items that were in the same local region at that same time (agents are encouraged to collect the oldest item in their region first).
>
> We have also included two hand-coded baselines to compare against (black dashed lines in Figures 3, 5, and 6). For the traffic domain, we have used traffic light controllers that use sensors to adapt to the traffic. These controllers were extensively optimized by Wu et al. (2017). For the warehouse domain, we hand-coded policies that follow the shortest path toward the oldest item in the region. Note that this is actually a very strong baseline.
>
> If the paper is accepted, we will use the extra page in the final version to extend the environment descriptions and the discussion of the experimental results.
>
> **Agent - simulator groupings:**
>
> Having one simulator per agent is certainly not a requirement. We do think that in some environments (including the two we explore here) you may obtain better results by grouping some of them together in the same simulator. In fact, one could potentially treat the agents in the same group/simulator as a single agent and train a policy to control all of them simultaneously. Note, however, that this is orthogonal to our work. In this paper, we are mainly concerned with computational speed-ups. That is why, in our experiments, we use one simulator per agent as this is the most computationally efficient way of factorizing the environment. However, we find this idea interesting and have added a discussion in Appendix F.2.
>
> **Non-improvement using recurrent neural networks:**
>
> We are not sure what result the reviewer is referring to. If the reviewer means the untrained-DIALS, (blue curves in Figure 3), then their intuition is correct. Since the AIPs are not trained, the local simulators produce unrealistic trajectories that are far from the ones we see in the true environment (global simulator). Hence, agents overfit to these unrealistic trajectories and even though they perform well in the untrained-DIALS, their policies perform poorly on the global simulator because they do not generalize well to the true dynamics.
>
> **Time index as input to the AIPs:**
>
> The time index is not given as input to the influence estimators (AIPs). We conducted some experiments and adding the time-index to the input does not seem to reduce the cross-entropy loss of the AIPs. Note that if the AIPs are RNNs they can potentially learn a time counter if this is important for predicting the influence.
>
> **Number of rollouts per training batch:**
>
> The number of rollouts (samples) per training batch is 32. This is shown in Table 6 Appendix H (batch size).
>
> **Limitations - tuning the simulation scheme:**
>
> We have added a new section in the appendix (Appendix C) where we discuss the limitations and provide insights on what are the requirements in terms of domain knowledge for building a simulator in practice.

---

> > ### Comment · Reviewer_EtVj · 2022-08-06
> > **Thanks!**
> >
> > Great, thanks to the authors for these clarifications. I was already strongly in favor of this paper being accepted and remain so after the above discussion.

---

### Official Review · Reviewer_yftC · 2022-07-13

**Rating:** 5
**Confidence:** 3
**Soundness:** 3 good
**Presentation:** 4 excellent
**Contribution:** 3 good

**Summary:**

This work focuses on factorizing large networked system simulators into smaller local simulators for training multi-agent reinforcement learning. The proposed method creates an influence augmented local simulator by training an influence predictor using data from the global simulator and applying this predictor to the local simulator.

**Questions:**

see above

**Limitations:**

yes

**Strengths And Weaknesses:**

Factorization is a common methodology in MARL training. This work proposes a different perspective which is factorizing the underlying simulator to improve scalability in the number of agents and the size of the simulated environment. The side effect of the method in mitigating the non-stationary issue is also worth further investigation. The manuscript is overall well written and easy to follow. A preliminary theoretical analysis is also provided.

The following are some of my questions and concerns:
- Factorizing the large simulator into smaller simulators involves running multiple processes of smaller simulators and training/executing the AIP, the overhead of such a method should be evaluated. The tradeoff information between the number of local simulators and additional memory and computation overhead would be helpful. It gives us a better sense in terms of what size of simulator should be partitioned and how many partitions it should have.
- The large simulator is factorized into many single-agent environments. To improve the training efficiency and resource utilization efficiency, it may be more beneficial to partition the large simulator into multiple multi-agent simulators. This should be considered in the evaluation section.
- Although periodic update of AIP with slower frequency seems to stabilize the training. I wonder if that would affect exploration of the MARL training. For instance, if there is an optimal joint state that is harder to achieve, would the slow update of AIP slows down the exploration and convergence?

---

> ### Author Response · Authors · 2022-08-02
> **Response**
>
> We thank the reviewer for their valuable and constructive feedback, which has helped us improve the paper. Please find the answers to your questions below.
>
> **Computation - memory tradeoff:**
>
> We agree with the reviewer that discussing the tradeoff between computation and memory usage is important. We have added a new Section in the appendix on the scope and limitations of the method where we discuss the tradeoffs between computational speedups vs memory (Appendix C last paragraph). We have also included a table in Appendix I that shows the peak memory usage of GS and DIALS for the two environments and the four scenarios. As for the computation overhead, the tables in Appendix H, which were already included in the original submission, show a breakdown of the runtimes.
>
> **Simulator partitioning - Number of regions vs Number of agents:**
>
> It is true that in certain applications, due to hardware limitations (e.g. not enough CPUs or memory available), it might be necessary to partition the simulator into fewer local regions than the number of agents in the environment. However, note that the needs (availability) of memory or CPUs are very domain specific and thus, it would be up to the system designer to choose the right configuration. When there are no constraints, the most efficient partitioning (in terms of compute) is to split the environment into the smallest possible local regions such that we can benefit from parallelization and readily scale up to larger environments. That is why in our experiments since we are investigating computational speedups compared to using the global simulator, we use one simulator for each agent. That said, we find the reviewer's point interesting and have included a discussion of the mentioned tradeoffs in Appendix F.2.
>
> **Effect of AIP training frequency on exploration and convergence:**
>
> The reviewer is right in that, in some environments, if the AIPs are not updated frequently enough, learning might be slowed down. However, in general, we believe that DIALS will be most effective in domains where the local regions are weakly coupled, and hence updating the AIPs very frequently would not be necessary. This is true for the two environments and four variants we explore here, where, in fact,  high-frequency updates are actually detrimental to learning (see Figure 4, and Figures 7 and 8 in Appendix E.2). The AIP training frequency is controlled by the hyperparameter F. Its effect on learning and convergence, is investigated in Section 5.3 (third paragraph).  We think that investigating how to adapt this approach to more strongly coupled domains is a very interesting question for future research and have mentioned it in the conclusion. We will also use part of the extra page in the final version to further discuss this idea if the paper is accepted.

---

### Meta-Review · Area_Chair_rtx6 · 2022-08-27

**Recommendation:** Accept
**Confidence:** Certain

**Metareview:**


All the reviewers agree that the paper present novel and interesting contributions. As such I recommend acceptance.

Please incorporate the last feedback from Reviewer iVuf.


**Award:**

No

---

### Decision · Program_Chairs · 2022-09-14

Accept